# GraphBench: Next-generation graph learning benchmarking

## Abstract

Machine learning on graphs has recently achieved impressive progress in various domains, including molecular property prediction and chip design. However, benchmarking practices remain fragmented, often relying on narrow, task-specific datasets and inconsistent evaluation protocols, which hampers reproducibility and broader progress. To address this, we introduce GraphBench, a comprehensive benchmarking suite that spans diverse domains and prediction tasks, including node-level, edge-level, graph-level, and generative settings. GraphBench provides standardized evaluation protocols—with consistent dataset splits and performance metrics that account for out-of-distribution generalization—as well as a unified hyperparameter tuning framework. Additionally, we benchmark GraphBench using message-passing neural networks and graph transformer models, providing principled baselines and establishing a reference performance.

## 1 Introduction

Machine learning on graphs using *graph neural networks* (GNNs), specifically *message-passing neural networks* (MPNNs) (Gilmer et al., 2017; Scarselli et al., 2009) and *graph transformers* (GTs) (Müller et al., 2024a), has become a cornerstone of modern machine learning research, spanning applications in drug design (Wong et al., 2023), recommender systems (Ying et al., 2018), chip design (Chen et al., 2025), and combinatorial optimization (Cappart et al., 2021). Despite these advances, benchmarking in the field remains highly fragmented. Narrow, task-specific datasets and inconsistent evaluation protocols limit reproducibility and hinder meaningful comparison across tasks and domains (Bechler-Speicher et al., 2025). As Bechler-Speicher et al. (2025) argue, current benchmarks tend to emphasize specific domains—such as 2D molecular graphs—while overlooking more impactful, real-world applications like relational databases, chip design, and combinatorial optimization. Many benchmark datasets fail to accurately reflect the complexity of real-world structures, resulting in inadequate abstractions and misaligned use cases. In addition, the field's heavy reliance on accuracy as a primary metric encourages overfitting, rather than fostering generalizable, robust models. As a result, these limitations have significantly hindered progress in graph learning.

**Shortcomings of current most-used graph learning datasets** Existing graph benchmarks have driven progress through standardized datasets and evaluations, but suffer from key limitations. TUDatasets (Morris et al., 2020) introduced many graph-level tasks, yet most are small, molecular, and lack unified metrics, hindering comparability. The *Open Graph Benchmark* (OGB) (Hu et al., 2020a) and its extensions (Hu et al., 2021) offer large-scale datasets; however, they focus heavily on molecules, require significant computational resources, and have unclear broader relevance. Other efforts, such as Dwivedi et al. (2020) and the *Long-range Graph Benchmark* (LRGB) (Dwivedi et al., 2022a), emphasize narrow prediction regimes, small-scale data, and restricted model sizes, with limited evaluation of out-of-distribution generalization. Benchmarks for graph generation remain underdeveloped (Bechler-Speicher et al., 2025), often relying on 2D molecular datasets with questionable practical value, synthetic data with limited applicability, and tasks that fail to address challenges such as generating large or structurally constrained graphs. Recent work has sought to formalize dataset quality assessment (Coupette et al., 2025) and expand node-level tasks with GraphLand (Bazhenov et al., 2025), but scalability and coverage gaps remain.

**Present work** We introduce GraphBench, a next-generation benchmarking suite addressing key shortcomings in graph learning evaluation. GraphBench spans diverse domains—algorithmic

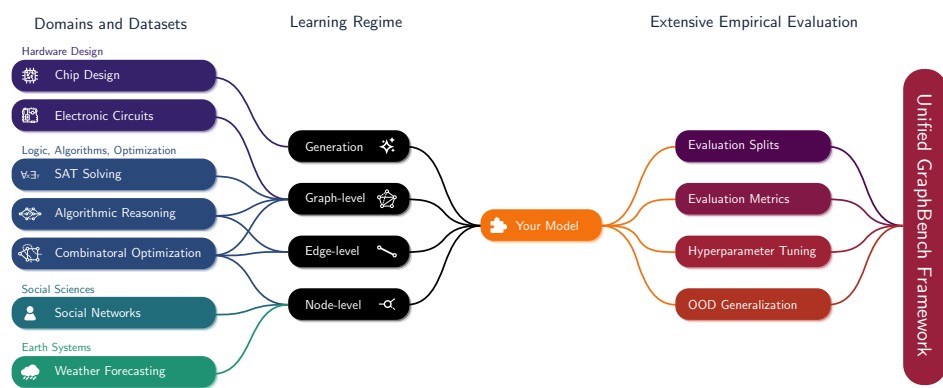

Figure 1: **Overview of the GraphBench framework.** GRAPHBENCH integrates diverse **domain datasets**—ranging from social sciences and hardware design to logic, optimization, and earth systems—into different **learning regimes** (node-, edge-, graph-level prediction and generative tasks). GRAPHBENCH offers **extensive empirical evaluation** via standardized splits, metrics, hyperparameter tuning scripts, and out-of-distribution (OOD) generalization tests, all within a **unified evaluation pipeline**.

reasoning, chip design, combinatorial optimization, SAT solver prediction, social networks, and weather forecasting—and supports node-, edge-, graph-level, and generative tasks. It provides standardized splits, domain-specific metrics, and hyperparameter tuning scripts, along with OOD evaluation for selected datasets to measure generalization. We benchmark modern MPNNs and graph transformers, offering strong baselines and insights into architectural trade-offs. *By unifying heterogeneous tasks under one framework,* GRAPHBENCH *enables reproducible, robust, and impactful graph learning research; see Figure 1.*

## 2 OVERVIEW OF GRAPHBENCH

The goal of GRAPHBENCH is to support the next generation of graph learning research by addressing key shortcomings of existing benchmarks. Here, we provide an overview of the design and features of GRAPHBENCH. GRAPHBENCH provides a unified framework for evaluating models across diverse graph tasks, domains, and data scales. Unlike previous benchmarks that focus narrowly on specific applications (e.g., molecular graphs or citation networks), with little (industrial) impact, GRAPHBENCH includes datasets from *social networks*, *chip design*, *combinatorial optimization*, *SAT solving*, and more.

GRAPHBENCH is designed around the following core principles.

1. **Diverse tasks and domains** GRAPHBENCH supports node-, edge-, and graph-level prediction, as well as generative tasks. Datasets span domains such as social networks, chip and analog circuit design, algorithm performance prediction, and weather forecasting. This diversity enables the evaluation of models in both traditional and emerging graph learning applications.

2. **Real-world impact** Many existing benchmark suites focus on small-scale graphs or domains that have limited applicability to current real-world challenges. In contrast, GRAPHBENCH emphasizes datasets and tasks that are more representative of modern real-world use cases, such as large-scale chip design, weather forecasting, and combinatorial optimization. By moving beyond the citation network- and molecular-dominated domains of prior benchmarks (e.g., OGB), GRAPHBENCH offers scenarios that better align with the needs of industry and emerging research areas.

3. **OOD generalization** Several datasets in GRAPHBENCH are explicitly split by time or problem size to test a model's ability to generalize beyond the training distribution. For instance, social network benchmarks assess future interaction prediction using only past data, while circuit benchmarks evaluate generalization across different circuit sizes.

4. **Meaningful evaluation** GRAPHBENCH provides evaluation splits, meaningful domain-specific evaluation metrics, and state-of-the-art hyperparameter tuning scripts. In addition, we offer out-of-distribution generalization evaluation for selected datasets, focusing on size generalization abilities, paving the way for graph foundation models.

5. **Task-relevant evaluation metric** GRAPHBENCH provides metrics tailored to each task, designed to capture the real-world performance of models. These measures extend beyond classical metrics such as MSE and accuracy, which may not accurately reflect practical utility in real-world scenarios.

**Benchmarks and evaluation**    To facilitate fair and reproducible comparisons, GRAPHBENCH offers a comprehensive benchmarking protocol for each dataset, including (1) a clearly defined prediction or generation task (e.g., node-, link-, or graph-level classification/regression, or graph generation), (2) a realistic and domain-appropriate data split strategy (e.g., temporal splits for social networks, cross-size splits for electronic circuits), (3) standardized input features and labels to ensure consistency across models, and (4) evaluation scripts implementing task-specific metrics (e.g., RMSE, ROC-AUC, closed gap) along with state-of-the-art hyperparameter tuning scripts. This standardized setup enables researchers to compare methods under realistic conditions while reducing the burden of dataset curation and preprocessing.

**Baselines and experimental evaluation**    To establish strong reference points, we evaluate a range of graph learning architectures, including MPNNs and graph transformers. All are evaluated across multiple random seeds to assess robustness. Our empirical analysis reveals persistent challenges for current methods, including coping with temporal distribution shifts and maintaining efficiency during training on graph-structured data.

**Software and accessibility**    GRAPHBENCH is released as an open-source, user-friendly PYTHON package designed for ease of adoption and extension. It provides streamlined data loaders and PYTORCH GEOMETRIC-based data objects, predefined training, validation, and test splits for each dataset, and ready-to-use state-of-the-art hyperparameter tuning scripts. Its modular design allows practitioners and researchers to rapidly prototype, benchmark, and deploy new graph learning models while adhering to a unified evaluation protocol; see Appendix A.1 for an example of using GRAPHBENCH's interface.

## 3    OVERVIEW OF GRAPHBENCH'S DATASETS

In the following, we provide an overview of GRAPHBENCH's domains and the provided datasets.

### 3.1    SOCIAL SCIENCES

The social sciences domain in GRAPHBENCH includes datasets that model human interactions and societal processes as graphs, where nodes represent individuals or entities and edges represent relationships or communications. These graphs facilitate evaluation of tasks such as link prediction, community detection, and temporal forecasting, providing valuable insights into real-world social systems.

#### 3.1.1    SOCIAL NETWORKS: PREDICTING VIRALITY ON BLUESKY

Social networks—graphs with users as nodes and interactions as edges—provide a structurally rich domain for graph-based machine learning (Newman, 2010). They exhibit hubs, peripheral nodes, clustered communities, and evolving, often directed edges. Dynamic user interests create temporal dependencies that shape representations and predictions. These properties make social networks a challenging benchmark—pushing models to handle edge directionality and structural diversity. Our benchmark predicts the number of engagements a user will receive on upcoming posts—a user-centric measure of near-term influence with applications in ranking users, trend detection, influencer marketing, and proactive moderation.

**Description of the learning task**    We build upon the publicly available BLUESKY dataset curated by Failla & Rossetti (2024) and cast the problem as node-wise regression to predict an aggregated statistic on the engagements a user will receive on their future posts. In our setting, time is discrete, with $t \in \mathbb{N}$ typically denoting a time step (in seconds). For any interval $\tau_{A,B} := (t_A, t_B] := \{t_A+1, \ldots, t_B\}$, we define the following.

- **Interaction graph** $G^\iota_{\tau_{A,B}} := (V, E^\iota_{\tau_{A,B}})$ where $V$, the node set, consists of social network users, and $E^\iota_{\tau_{A,B}} \subseteq V \times V$ is the set of *directed* interactions of kind $\iota$ observed in $\tau_{A,B}$. Each edge $e = (u, v)$ indicates at least one $\iota$-interaction ($\iota \in \{\text{quotes}, \text{replies}, \text{reposts}\}$) between users $u$ and $v$ in interval $\tau_{A,B}$.

- **Content** $T_{\tau_{A,B}}^{(u)} := \{\mathbf{c}_{u,p_t} \mid t \in \tau_{A,B}\}$ collects the textual content for post $p$ authored by user $u \in V$ at any $t \in \tau_{A,B}$ (i.e., $\mathbf{c}_{u,p_t}$). As we will detail next, we consider aggregated vectorial representations of these sets of post texts, obtained via a pretrained language model.

- **Engagements** $E_{\tau_{A,B}}^{\kappa,(u)} := \{e_{u,p_t}^{\kappa} \mid t \in \tau_{A,B}\}$ collects the number of engagements $\kappa \in \{\text{likes, replies, reposts}\}$ received by posts $p$ authored by $u$ at $t \in (t_A, t_B]$. We define an aggregate statistic $y_{\tau_{A,B}}^{\kappa,(u)} = \text{median}\left(E_{\tau_{A,B}}^{\kappa,(u)}\right)$.[1]

Given timestamps $t_0 < t_1 < t_2$, user $u$ and target engagement $\kappa$, the learning task is to predict value $y_{\tau_{1,2}}^{\kappa,(u)}$ given the overall interaction graph $G_{\tau_{0,1}}^{\iota}$ and post contents $\{T_{\tau_{0,1}}^{(u)} \mid u \in V\}$. We assess model performance using the Mean Absolute Error (MAE), coefficient of determination $R^2$, and the Spearman correlation $\sigma$, where the last two are calculated specifically for each target engagement $\kappa$. We note that $\sigma$ is relevant to us since it measures how well predictions preserve the relative ordering of users by future engagements, a key criterion when ranking influence in heavy-tailed social networks, not necessarily captured by MAE and $R^2$.

Table 1: **Social Networks.** Performance metrics for each model on the different datasets. Test MAE, $R^2$, and Spearman correlation are reported.

| Dataset | Model | MAE | $R^2_{\text{likes}}$ | $R^2_{\text{replies}}$ | $R^2_{\text{reposts}}$ | $\sigma_{\text{likes}}$ | $\sigma_{\text{replies}}$ | $\sigma_{\text{reposts}}$ |
|---|---|---|---|---|---|---|---|---|
| quotes | DeepSets | $0.810 \pm 0.005$ | $0.140 \pm 0.002$ | $0.102 \pm 0.002$ | $0.138 \pm 0.002$ | $0.307 \pm 0.004$ | $0.178 \pm 0.002$ | $0.334 \pm 0.003$ |
| | GNN | $0.768 \pm 0.002$ | $0.165 \pm 0.002$ | $0.134 \pm 0.002$ | $0.175 \pm 0.002$ | $0.330 \pm 0.003$ | $0.192 \pm 0.002$ | $0.337 \pm 0.022$ |
| | MLP | $0.784 \pm 0.001$ | $0.145 \pm 0.001$ | $0.107 \pm 0.001$ | $0.141 \pm 0.001$ | $0.308 \pm 0.003$ | $0.176 \pm 0.002$ | $0.335 \pm 0.002$ |
| replies | DeepSets | $0.789 \pm 0.033$ | $0.086 \pm 0.033$ | $0.061 \pm 0.024$ | $0.104 \pm 0.014$ | $0.253 \pm 0.003$ | $0.130 \pm 0.001$ | $0.240 \pm 0.006$ |
| | GNN | $0.694 \pm 0.002$ | $0.158 \pm 0.003$ | $0.119 \pm 0.002$ | $0.159 \pm 0.002$ | $0.258 \pm 0.009$ | $0.132 \pm 0.006$ | $0.247 \pm 0.011$ |
| | MLP | $0.725 \pm 0.004$ | $0.131 \pm 0.003$ | $0.087 \pm 0.003$ | $0.122 \pm 0.003$ | $0.249 \pm 0.004$ | $0.127 \pm 0.005$ | $0.243 \pm 0.003$ |
| reposts | DeepSets | $0.918 \pm 0.013$ | $0.049 \pm 0.005$ | $0.031 \pm 0.012$ | $0.050 \pm 0.009$ | $0.229 \pm 0.004$ | $0.129 \pm 0.007$ | $0.205 \pm 0.009$ |
| | GNN | $0.832 \pm 0.009$ | $0.131 \pm 0.017$ | $0.111 \pm 0.022$ | $0.128 \pm 0.008$ | $0.284 \pm 0.044$ | $0.143 \pm 0.032$ | $0.260 \pm 0.051$ |
| | MLP | $0.874 \pm 0.003$ | $0.087 \pm 0.002$ | $0.051 \pm 0.003$ | $0.066 \pm 0.002$ | $0.234 \pm 0.008$ | $0.123 \pm 0.006$ | $0.206 \pm 0.010$ |

**Experimental evaluation and baselines** Table 1 shows a clear, consistent ordering from DeepSets to MLP to GNN, across all interaction types (quotes, replies, reposts) in terms of lower MAE, higher $R^2$, and Spearman ($\sigma$). These results suggest that *local, directed relational structures* are informative for predicting engagements in a short future time span, as both DeepSets and MLPs, which ignore graph structure, lag behind GNNs. Unlike MLPs, DeepSets consider *global* feature information at the level of the entire network, which appears to be detrimental. We also note that the absolute values of $R^2$ and $\sigma$ remain relatively low, indicating that the proposed tasks are far from being solved and leaving room for meaningful gains through more advanced, task-specific GNNs.

### 3.2 HARDWARE DESIGN

The hardware design domain in GRAPHBENCH encompasses datasets that represent electronic circuits through graph structures, where nodes correspond to elements such as transistors, resistors, or logic gates, and edges denote their electrical connections. These graphs are well-suited for evaluating models on tasks such as circuit optimization and performance prediction, all of which are critical for advancing efficient and reliable hardware development.

#### 3.2.1 CHIP DESIGN: LEARNING TO GENERATE SMALL CIRCUITS

A central challenge in Boolean circuit synthesis is ensuring optimized circuits remain functionally equivalent to the original. Regardless of transformations aimed at reducing gate count, delay, power, or area, outputs must match for all inputs. This makes the task delicate—small changes can alter behavior. Finding a minimal circuit is NP-hard, making synthesis a core computational challenge. Classical tools like ABC (Brayton & Mishchenko, 2010) use heuristics to generate near-optimal circuits efficiently. Recently, learning-based methods have emerged as alternatives, using data-driven models to capture structural patterns and generalize across designs. Boolean circuits' natural representation as directed acyclic graphs makes graph-based generative learning particularly promising.

---

[1]We observed the median be a more robust statistic w.r.t., e.g., the mean, in this context.

**Description of the learning task**   Given a set of $N \in \mathbb{N}$ Boolean functions, $\{f_i\}_{i=1}^N$, $f_i \colon \{0,1\}^{n_i} \to \{0,1\}^{m_i}$, the goal of Boolean circuit synthesis is to construct, for each function, a logic circuit $C_i = (V_i, E_i)$ with $n_i$ input and $m_i$ output nodes that satisfies the following conditions.

1. **Logical correctness (constraint)** For all $\boldsymbol{x} \in \{0,1\}^{n_i}$, the output of the circuit must match the output of $f_i$, i.e., $C_i(\boldsymbol{x}) = f_i(\boldsymbol{x})$, where $C_i(\boldsymbol{x})$ denotes the vector of output values produced by the circuit on input $\boldsymbol{x}$. In that case, $C_i$ is said to be *logically equivalent* to $f_i$, which we denote $C_i \equiv f_i$.

2. **Structural efficiency (objective)** For each $f_i$, among all circuits $C = (V, E)$ satisfying logical correctness, select a circuit that optimizes a predefined circuit score function $\mathrm{s}(C)$, where $\mathrm{s} \colon \mathcal{C}_{f_i} \to \mathbb{R}_{\geq 0}$, which measures structural properties of the circuit, such as the number of gate nodes,

$$\arg\max_{C \in \mathcal{C}_{f_i}} \mathrm{s}(C),$$

where $\mathcal{C}_{f_i} = \{C \mid C \equiv f_i\}$ is the set of all circuits implementing $f_i$. The score function should favor AIGs with fewer internal nodes while penalizing incorrect solutions. Therefore, we define score over the set of functions $\{f_i\}_{i=1}^N$ as

$$\mathrm{score}(\tilde{C}) := \frac{100}{N} \sum_{i=1}^N \frac{\# C_i^{\mathrm{ref}}}{\# \tilde{C}_i} \cdot \mathbb{1}_{\tilde{C}_i \equiv f_i}, \tag{1}$$

where $\tilde{C}$ is the set of generated AIGs, $\# \tilde{C}_i$ and $\# C_i^{\mathrm{ref}}$ are the number of internal nodes of the $i$-th generated circuit and of the corresponding baseline AIG generated by ABC, $\mathbb{1}_{\tilde{C}_i \equiv f_i} = 1$ if $\tilde{C}_i$ is logically equivalent to $f_i$, and 0 otherwise.

**Experimental evaluation and baselines**   We report results obtained using four standard methods within ABC (see Table 7), where the score is defined in Equation (1). Notably, we do not include any learning-based baselines for this dataset. While numerous graph generative approaches exist (e.g., Vignac et al. (2023)), most are designed for undirected graphs and are thus unsuitable for DAGs. Moreover, our experiments with two DAG-specific generative methods, LayerDAG (Li et al., 2025a) and Directo (Carballo-Castro et al., 2025), indicate that they fail to produce circuits functionally equivalent to the target truth table. This suggests a new and unexplored direction for the graph generation community: developing conditional DAG generative models that enforce strict constraints, such as functional equivalence with a given truth table.

### 3.2.2 ELECTRONIC CIRCUITS: PREDICTING VOLTAGE CONVERSION RATIO

Analog circuit design remains critical but labor-intensive in *electronic design automation* (EDA) (Zhang et al., 2019; 2020; Zhao & Zhang, 2020b). Unlike mature digital flows, analog design still relies on expert intuition due to sensitivity to device variations and complex constraints. This is especially true for custom power-converter design, where topology generation and parameter tuning must balance conflicting metrics. Even with a modest number of components, hundreds of thousands of topologies are possible; yet, new converter families are mostly handcrafted. Automating this process could accelerate design cycles and expand the exploration of novel architectures. Learning-based methods offer promise as surrogate models that predict performance (Fan et al., 2021; 2024b), enabling faster search over vast spaces. Power converters' natural graph structure makes them an ideal benchmark, stressing robustness to structural sensitivity and coping with extreme class imbalance.

**Description of the learning task**   Given a graph $G$, encoding a power converter, the prediction task is to estimate two continuous performance metrics: (1) the *voltage conversion ratio* (output-to-input voltage) and (2) the *power conversion efficiency* (fraction of input power delivered to the load). Performance is evaluated via the *relative squared error*, reported separately for each target,

$$\mathrm{RSE}(y, \hat{y}) := \frac{\sum_{i=1}^N (y_i - \hat{y}_i)^2}{\sum_{i=1}^N (y_i - \bar{y})^2}, \tag{2}$$

where $y_i$ is the ground-truth value from high-fidelity simulation, $\hat{y}_i$ the model prediction, $\bar{y}$ the mean of $\{y_i\}_{i=1}^N$, and $N$ the number of evaluation samples.

**Experimental evaluation and baselines** We evaluate four standard methods, including the Graph Transformer and MPNN-based models (see Table 8), using the score defined in Equation (2). Experiments are conducted on datasets containing 5, 7, and 10-component circuits. For each dataset, models are trained from scratch and evaluated on circuits of the same size. Table 8 reports the relative squared error RSE for efficiency and output voltage prediction, comparing the GT with baseline models (GCN, GAT, and GIN) as circuit complexity increases. Results confirm that the GT consistently achieves higher prediction accuracy than baseline models, not only on smaller circuits but also across datasets with varying circuit sizes and prediction targets. The rise in RSE for larger circuits is mainly due to the reduced availability of training data from costly high-fidelity simulations, combined with the combinatorial growth of circuit space, which makes learning increasingly difficult for all models.

## 3.3 REASONING AND OPTIMIZATION

This GRAPHBENCH domain targets graph learning tasks involving complex reasoning and hard optimization, from theorem proving and hardware verification to logistics and network design. Models must capture fine-grained structural dependencies and generalize across instances. We highlight three problem families: (1) *SAT solver selection and performance prediction*, which forecasts solver behavior or picks the best solver per instance; (2) *combinatorial optimization*, including NP-hard problems like maximum independent set, max-cut, and graph coloring; and (3) *algorithmic reasoning*, where models approximate outputs of polynomial-time graph algorithms. Benchmarks evaluate predictive accuracy, robustness, scalability, and (out-of-distribution) generalization.

### 3.3.1 SAT SOLVING: ALGORITHM SELECTION AND PERFORMANCE PREDICTION

SAT is one of the first problems proven NP-complete (Cook, 1971) and remains theoretically significant. Beyond its role in complexity theory, it has many applications, including hardware/software verification (Biere et al., 2009), automated planning (Kautz & Selman, 1996), and operations research (Gomes et al., 2008). SAT instances can be expressed as graphs, reflecting the permutation-invariant structure from the commutativity and associativity of conjunction and disjunction.

**Description of the learning task** We provide two learning tasks. *Performance prediction* is a regression problem whose goal is to predict the computation time of SAT solvers on unseen instances. Such models are used in applications, such as algorithm selection (Rice, 1976) and algorithm configuration, e.g., (Lindauer et al., 2022). *Algorithm selection* is a multi-class classification problem that aims to select the best algorithm for a given instance. For the SAT benchmarks, each instance is provided with three graph-based representations VG, VCG, and LCG. In addition, we utilized SATZILLA features; see Appendix B.3.1 for details. To create the dataset, we constructed, to the best of our knowledge, the largest algorithm selection and performance prediction dataset for SAT solvers, with more than 100 000 instances. We gathered eleven complementary SAT solvers from the 2023 and 2024 SAT Competitions. Additional details on the tasks, metrics, and data collection and generation are available in Appendix B.3.1. We provide three datasets, SMALL, which includes only small formulae with up to 3 000 variables and 15 000 clauses, MEDIUM, which consists of the small formulae with additional medium size formulae with size of up to 20 000 variables and 80 000 clauses and LARGE, which includes all formulae. This way, we provide datasets with different hardness levels, as the size of the formula can cause hurdles related to training time and GPU memory. While the LARGE and MEDIUM datasets are unlikely to be accessible to current hardware, we provide it to allow the benchmark to stay relevant in the long term.

Table 2: **SAT Solving.** RMSE ($\log_{10}$ time) for performance prediction on SMALL SAT instances. We report results on three different graph representations (VG, VCG, LCG) and using the hand-crafted SATzilla features. Lower is better.

| Solver | Method | VG | VCG | LCG | SATZILLA |
|--------|--------|------|------|------|----------|
| | GIN | $1.36 \pm 0.15$ | $1.10 \pm 0.03$ | $1.49 \pm 0.02$ | – |
| KISSAT | RF | – | – | – | $0.61 \pm 0.00$ |
| | XGB | – | – | – | $0.63 \pm 0.00$ |

**Experimental evaluation and baselines** In addition to graph-based learning approaches, we compare our method with traditional methods that rely on handcrafted features, specifically, the SATzilla 2024 feature

set (Shavit & Hoos, 2024). Due to space constraints, we report results for performance prediction on the winner of the 2024 SAT Competition—KISSAT. Additional results, including performance prediction for other solvers and algorithm selection, are provided in Appendix B.3.1. As our evaluation metric, we use the RMSE of log-scaled $PAR_{10}$ scores, i.e., the solver computation time, with timeouts penalized at $10\times$ the cutoff time on the small formulae benchmark in Table 2. The results show that traditional empirical performance predictors, including random forest and XGBoost, which utilize handcrafted features, outperform GNN-based approaches by a wide margin. We also observe substantial differences between graph representations: while LCG is commonly used as GNN input in SAT-solving contexts (Selsam et al., 2019; Tönshoff & Grohe, 2025), the more compact VCG and VG representations yield better performance. This highlights that the choice of input graph can strongly influence predictive accuracy in SAT-solving tasks. For the algorithm selection task, we observe a similar trend: methods based on hand-crafted features consistently outperform graph learning approaches. Full results for this task are included in Appendix B.3.1.

### 3.3.2 COMBINATORIAL OPTIMIZATION: LEARNING TO PREDICT OPTIMAL SOLUTION OBJECTIVES

In *combinatorial optimization* (CO), the goal is to find a solution from a discrete set that optimizes a given objective under constraints. CO underpins applications such as vehicle routing, scheduling, and resource allocation (Paschos, 2014). CO problems are particularly well-suited for learning-based research: their hardness (Karp, 1972) motivates GNNs as efficient surrogates for solving large instances. Many are naturally graph-structured, requiring expressive architectures (Chen et al., 2023; 2024). Additionally, large synthetic datasets can be generated for robust benchmarking. Exact solvers can yield optimal solutions for small cases and strong approximations for larger ones, supporting both supervised training and generalization studies. The objective value provides a clear quantitative signal, making CO an ideal testbed for advancing graph learning.

**Description of the learning task**   We include tasks for both supervised and unsupervised learning settings. In the supervised case, the goal is to train a graph model to predict the optimal objective value for a given CO instance. Predicting values rather than explicit solutions avoids the challenge of defining a loss when multiple optima exist, enabling consistent training and evaluation. However, supervised training depends on solver-generated optimal solutions, which become intractable for large instances. This motivates the unsupervised setting, which is valuable when ground-truth solutions are unavailable or expensive. We provide a differentiable surrogate loss and problem-specific decoders; the model outputs a score per variable indicating its likelihood of belonging to the solution. At test time, scores are decoded into feasible solutions whose quality is measured by the resulting objective. This approach, common in CO graph learning (Karalias & Loukas, 2020; Min et al., 2022; Wenkel et al., 2024), avoids costly ground-truth computation and directly targets solving CO problems. We consider three CO problems in both settings—*maximum independent set*, *maximum cut*, and *graph coloring*—on three synthetic graph families: RB (Xu et al., 2007), *Erdős-Rényi* (ER) (Erdős & Rényi, 1960), and *Barabási-Albert* (BA) (Albert & Barabási, 2002). Task details appear in Appendix B.3.2.

**Experimental evaluation and baselines**   We evaluate our model using RWSE (Dwivedi et al., 2022b) and degree features on the supervised MIS objective prediction task. We train the model with MAE loss and report the test loss. Baselines include GIN, GT, MLP, and DeepSet. As shown in Table 14, GIN achieves the strongest performance due to the inductive bias of MPNNs. DeepSet generally outperforms MLP, except on the ER dataset. GT performs poorly on some datasets, likely due to training instability. Table 15 shows the baseline results for unsupervised CO. We report the average objective value of solutions obtained by running the decoder on the model's output scores, for problem instances in the test set. Out of the four baselines, GIN performs best across all three CO problems. As expected, the graph-based models GIN and GT perform better than MLP and DeepSet. Note that in general, learning-based methods currently cannot compete with exact solvers and some hand-crafted heuristics in terms of solution quality, but are often much quicker (Zhang et al., 2023; Sanokowski et al., 2024; Sun & Yang, 2023).

### 3.3.3 ALGORITHMIC REASONING: LEARNING TO SIMULATE ALGORITHMS

In addition to SAT, many real-world tasks depend on the efficient solution of graph algorithms. With the intersection of algorithms and neural networks being actively explored across domains (Estermann et al., 2024; Fan et al., 2024a; Kaiser & Sutskever, 2016; Zaremba & Sutskever, 2014), our goal is to provide dedicated datasets for graph algorithmic reasoning. Prior work (Velickovic et al., 2020b; Xu et al., 2020; Bounsi et al., 2024) and benchmarks such as CLRS (Velickovic et al., 2022) have demonstrated that GNN

architectures can achieve strong performance on graph problems when provided with hints and graph invariants—tasks that require substantial theoretical expressiveness from graph learning models (Arvind et al., 2020). To broaden this research area, we contribute large-scale datasets for several common graph algorithm tasks, expanding the landscape of neural algorithmic reasoning benchmarks.

**Description of the learning task**    We include graph data for eight classic graph algorithms: minimum spanning tree, bridge finding, topological sorting, maximum flow, Steiner tree computation, topological sorting, maximum matching, and maximum clique finding, following the descriptions in the CLRS textbook (Cormen et al., 2009). While SAT solving is generally NP-complete, we focus here on polynomial-time algorithms to benchmark neural algorithmic reasoning. Each dataset consists of initial graph data with node and edge features, along with ground-truth results computed using NETWORKX's implementation of the respective algorithms (Hagberg & Conway, 2020). In contrast to prior benchmarks (Velickovic et al., 2022; Ibarz et al., 2022), we do not provide hints during training, allowing us to directly evaluate whether a model can emulate a specific algorithm from raw input. To enable size-generalization testing, we also provide additional data with larger graph instances reserved for evaluation. For each dataset, the goal is to approximate the result produced by the corresponding algorithm.

**Experimental evaluation and baselines**    Each dataset specifies the loss function and evaluation target for node-, edge-, or graph-level prediction, using metrics such as MAE and F1 score. Every graph instance is sampled uniformly at random from the generation methods listed in Appendix B.3.3, with test graphs drawn with increased node counts to assess size generalization at test time. In Appendix B.3.3, we report baseline results across all tasks, comparing GNN and graph transformer models. Both baselines achieve their best performance on bridge finding and minimum spanning tree computation, while Steiner tree, max clique, and max matching computation remain more challenging.

## 4 EARTH SYSTEMS

The earth systems domain in GRAPHBENCH includes graph learning tasks built from geospatial, environmental, and climate data. Nodes represent locations, sensors, or regions, and edges capture spatial or physical relationships. These tasks are key for applications such as weather forecasting, climate impact assessment, and resource management. They test whether graph-based models can integrate heterogeneous features, capture long-range dependencies, and generalize across time and space for robust environmental decision-making.

### 4.1 WEATHER FORECASTING: MEDIUM-RANGE ATMOSPHERIC STATE PREDICTION

Weather forecasting has long been critical for agriculture, energy, and public safety (Diehl et al., 2013; Ramar & Mirnalinee, 2014). Traditional *numerical weather prediction* (NWP) is accurate but relies on computationally expensive physics-based simulations (Bauer et al., 2015). Recent machine learning methods offer faster, scalable alternatives that can outperform NWP for medium-range forecasts and extreme events—while improving uncertainty quantification (Price et al., 2023). Earth system data are sparse, irregular, and interconnected (Reichstein et al., 2019), with interactions spanning scales from local convection to planetary waves (Bauer et al., 2015). Graph-based models naturally capture these multi-scale, non-Euclidean dependencies, enabling flexible representations of local and global atmospheric interactions.

**Description of the learning task**    The objective of the task is to model medium-range weather evolution by predicting the residual change in the atmospheric state over a fixed time horizon of twelve hours. Specifically, given an initial snapshot of the current atmospheric state, the model forecasts the twelve-hour future change in meteorological variables. The training objective is a spatially and variable-weighted *mean-squared error* for the twelve-hour-ahead prediction. For each verification time $d \in D$, the model ingests $x_{d-2}$ and predicts a residual change $\Delta x$, yielding $\hat{x}^d = x_{d-2} + \Delta x$. The loss compares $\hat{x}^d$ with $x^d$. The loss function thus corresponds to

$$\mathcal{L}(x^d, \hat{x}^d) = \frac{1}{|D|\,|G| \sum_{j \in J} |L_j|} \sum_{d \in D} \sum_{i \in G} \sum_{j \in J} \sum_{\ell \in L_j} a_i\, w_j\, s_{j,\ell} \left( \hat{x}^d_{i,j,\ell} - x^d_{i,j,\ell} \right)^2,$$

where $D$ is the set of forecast date-times, $G$ the grid cells, $J$ the variables, $L_j$ the pressure levels for variable $j$, $a_i$ are mean-normalized latitude weights, $w_j$ are variable weights, $s_{j,\ell}$ are the level weights and $P_\ell$ are the pressure levels.

**Experimental evaluation and baselines** For the evaluation, we report an unweighted mean-squared error for each surface variable $j$,

$$\text{MSE}_j(x^d, \hat{x}^d) = \frac{1}{|D|\,|G|\,|L_j|} \sum_{d \in D} \sum_{i \in G} \sum_{\ell \in L_j} \left( \hat{x}^d_{i,j,\ell} - x^d_{i,j,\ell} \right)^2,$$

and an average unweighted mean-squared error over the pressure levels 500, 700 and 850 that allow a direct comparison to existing weather models like Persistence and GraphCast (Lam et al., 2023). An overview is compiled in Table 3. A more detailed evaluation can be found in Appendix B.4.1.

Our baseline establishes a precise and reproducible lower bound for medium-range weather forecasting. While it does not reach the performance of Persistence or specialized systems like GraphCast, this is expected given their simpler architecture, shorter training, and deliberately straightforward setup. Importantly, the purpose of this baseline is not to approach state-of-the-art performance. Its main contribution lies in providing a transparent and reproducible lower bound against which future graph learning methods can be compared.

Table 3: **Weather Forecasting.** Mean squared error for each weather variable across selected pressure levels. Persistence denotes a basic weather forecasting model, whereas GraphCast denotes the model used by Lam et al. (2023). Table formatting k indicates a multiplicative factor of $10^3$ while n indicates $10^{(-9)}$.

| | Surface | | | | | Average over 500, 700 and 850 | | | | | |
|---|---|---|---|---|---|---|---|---|---|---|---|
| Method | 2T | 10U | 10V | MSL | TP | T | U | V | Z | Q | W |
| GT | 7.572 | 5.072 | 6.102 | 102.551k | 0.009 | 2.605 | 10.964 | 17.816 | 63.155k | 0.010 | 0.024 |
| Persistence | 7.123 | 2.166 | 3.266 | 60.056k | 714.517n | 1.174 | 4.875 | 8.769 | 37.954k | 223.608n | 4.370 |
| GraphCast | 0.068 | 0.012 | 0.013 | 240.832 | 52.377n | 0.007 | 0.031 | 0.034 | 145.356 | 3.092n | 0.034 |

## 5 FORMAT, LICENSING, AND LONG-TERM ACCESS

All GRAPHBENCH datasets are distributed as PYG objects with graph data, and raw data is also available in HDF5 for long-term compatibility. For algorithmic reasoning, weather forecasting, combinatorial optimization, and SAT solving, datasets can be regenerated using the provided code. Data is released under open licenses: Apache 2.0 for most datasets (including WeatherBench2), MIT for SAT instances, and GPL for AClib. Regarding social networks, we report that the original BlueSky data are hosted at `https://zenodo.org/records/11082879` under the Creative Commons Attribution 4.0 International License. The owners of such data may update these records to comply to users' "Right to Erasure"; we implement a system to update our derived dataset consistently. GRAPHBENCH is continuously updated with new datasets, and data is hosted on institutional servers and public repositories for persistent access.

## 6 CONCLUSION AND OUTLOOK

In this work, we introduce GRAPHBENCH, a next-generation benchmarking suite that addresses the fragmentation and limitations of existing graph learning benchmarks. By unifying diverse domains—social sciences, hardware design, reasoning and optimization, and earth systems—into a single standardized framework, GRAPHBENCH enables fair and reproducible comparisons across a wide range of graph tasks. Our benchmarks cover node-, edge-, and graph-level prediction as well as generative settings, with realistic splits, relevant metrics, and explicit tests for out-of-distribution generalization. Comprehensive baselines with modern MPNNs and graph transformers provide strong reference points for future work. Results reveal persistent challenges—including temporal distribution shifts, scaling to large graphs, and capturing domain-specific structures. By offering principled datasets, consistent evaluation, and robust baselines, GRAPHBENCH aims to drive the development of more generalizable and practically relevant graph learning methods.

**Vision for GRAPHBENCH** *Looking ahead, we envision* GRAPHBENCH *as a living benchmark—continually expanding to include new domains, tasks, and evaluation paradigms, and serving as a catalyst for progress in both fundamental research and impactful real-world applications of graph machine learning.* In addition, we aim to extend GRAPHBENCH so that it offers support for training the next-generation of multi-modal graph foundation models.

## 7 REPRODUCIBILITY STATEMENT

We provide code for reproducing our experiments in an anonymous GitHub repository: `https://anonymous.4open.science/r/graphbench-7DC1`. Since we introduce new large-scale datasets, anonymous access cannot be granted during the review process. However, as noted in Section 5, these datasets will be publicly released in a user-friendly format, enabling full reproducibility. Detailed dataset descriptions, including generation procedures, are given in Appendix B, and additional information about our experimental setup is provided in Appendix D.

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

# A  APPENDIX

## A.1  EXAMPLE USAGE OF GRAPHBENCH

We provide a unified and easy-to-use interface for GRAPHBENCH datasets. Inspired by benchmarks such as OGB (Hu et al., 2020a) and LRGB (Dwivedi et al., 2022a), we utilize wrapper methods to handle dataset loading. Unlike these benchmarks, however, GRAPHBENCH offers a single loading function (`graphbench.loader`) for all datasets, independent of task type. Each wrapper includes the necessary precomputations and comes with predefined splits, enabling direct use in downstream tasks. For seamless integration, all datasets are fully compatible with PYTORCH GEOMETRIC through its `InMemoryDataset` interface.

Beyond loading, GRAPHBENCH provides utilities for experimentation. Given a model, hyperparameter optimization can be run directly via `graphbench.optimize`, which implements the optimization algorithms described in Appendix D. For evaluation, `graphbench.evaluator` reproduces the settings used in our baselines, though users are free to integrate their own pipelines. Since specific datasets require specialized architectures (e.g., for graph generation), we also supply example architectures and usage guidelines. Finally, Listing 1 presents pseudocode illustrating how to train a predefined model using GRAPHBENCH.

```
1  import graphbench
2
3  model = #your torch model
4  dataset_name = #name of the task
5  pre_filter = #PyTorch Geometric filter matrix
6  pre_transform = #PyTorch Geometric-like transform during loading
7  transform = #PyTorch Geometric-like transform at computation time
8
9  #load a GraphBench dataset and get splits
10 dataset = graphbench.loader(dataset_name, pre_filter, pre_transform,
      transform)
11
12 #optimize your model
13 opt_model = graphbench.optimize(model, dataset['train'], dataset['valid'
      ])
```

Table 4: Summary of currently available GRAPHBENCH datasets.

| Category | Name | Node Feat. | Edge Feat. | Directed | Hetero | #Tasks | Split Scheme | Split Ratio | Task Type | Metric |
|---|---|---|---|---|---|---|---|---|---|---|
| Social networks | BlueSky – quotes | ✓ | – | ✓ | – | 3 | Temporal | (55/15)/15/15 | Node regression | MAE / $R^2$ / Spearman corr. |
| | BlueSky – replies | ✓ | – | ✓ | – | 3 | Temporal | (55/15)/15/15 | Node regression | MAE / $R^2$ / Spearman corr. |
| | BlueSky – reposts | ✓ | – | ✓ | – | 3 | Temporal | (55/15)/15/15 | Node regression | MAE / $R^2$ / Spearman corr. |
| Chip design | AIG | ✓ | ✓ | ✓ | – | 1 | Fixed | 80/10/10 | Generation | Ad-hoc Score |
| Electronic circuits | 5 Components | ✓ | – | – | – | 1 | Random | 70/10/20 | Regression | RSE |
| | 7 Components | ✓ | – | – | – | 1 | Random | 70/10/20 | Regression | RSE |
| | 10 Components | ✓ | – | – | – | 1 | Random | 70/10/20 | Regression | RSE |
| SAT Solving | SMALL | – | – | – | ✓ \ – | 5 | Fixed | 80/10/10 | Classification / Regression | Closed Gap / RMSE |
| | MEDIUM | – | – | – | ✓ \ – | 5 | Fixed | 80/10/10 | Classification / Regression | Closed Gap / RMSE |
| | LARGE | – | – | – | ✓ \ – | 5 | Fixed | 80/10/10 | Classification / Regression | Closed Gap / RMSE |
| Combinatorial optimization | RB – Small | – | – | – | – | 6 | Fixed | 70/15/15 | Regression / Unsupervised | MAE / CO |
| | RB – Large | – | – | – | – | 6 | Fixed | 70/15/15 | Regression / Unsupervised | MAE / CO |
| | ER – Small | – | – | – | – | 6 | Fixed | 70/15/15 | Regression / Unsupervised | MAE / CO |
| | ER – Large | – | – | – | – | 6 | Fixed | 70/15/15 | Regression / Unsupervised | MAE / CO |
| | BA – Small | – | – | – | – | 6 | Fixed | 70/15/15 | Regression / Unsupervised | MAE / CO |
| | BA – Large | – | – | – | – | 6 | Fixed | 70/15/15 | Regression / Unsupervised | MAE / CO |
| Algorithmic reasoning | Topological Sorting | ✓ | ✓ | ✓ | – | 3 | Fixed | 98/1/1 | Node regression | MAE |
| | MST | – | ✓ | – | – | 3 | Fixed | 98/1/1 | Edge classification | Accuracy / F1 |
| | Bridges | – | – | – | – | 3 | Fixed | 98/1/1 | Edge classification | Accuracy / F1 |
| | Steiner Trees | ✓ | ✓ | – | – | 3 | Fixed | 98/1/1 | Edge classification | Accuracy / F1 |
| | Max Clique | – | – | – | – | 3 | Fixed | 98/1/1 | Node classification | Accuracy / F1 |
| | Flow | ✓ | ✓ | ✓ | – | 3 | Fixed | 98/1/1 | Regression | MAE |
| | Max Matching | – | ✓ | – | – | 3 | Fixed | 98/1/1 | Edge classification | Accuracy / F1 |
| Weather forecasting | ERA5 64x32 | ✓ | ✓ | ✓ \ – | – | 1 | Temporal | 86/5/9 | Node regression | MSE |

```
14
15  #use graphbench evaluator
16  results = graphbench.evaluator(dataset_name, opt_model, dataset['valid'],
        dataset['test'])
```

Listing 1: PYTORCH-like pseudocode for GRAPHBENCH showcasing the usage of the dataset loader and application to downstream tasks.

Table 5: Statistics of currently-available GRAPHBENCH datasets.

| Category | Name | #Graphs | Avg. #Nodes | Avg. #Edges | Avg. Deg. |
|---|---|---|---|---|---|
| Social networks | BlueSky – quotes | 1 | 289 136 / 286 425 / 311 272 | 3 075 967 / 3 815 996 / 4 525 872 | 10.64 / 13.32 / 14.54 |
| | BlueSky – replies | 1 | 470 816 / 464 867 / 569 424 | 9 287 565 / 10 769 386 / 12 318 196 | 19.73 / 23.17 / 21.63 |
| | BlueSky – reposts | 1 | 498 012 / 508 818 / 580 112 | 12 049 951 / 14 658 114 / 17 261 865 | 24.20 / 28.81 / 29.76 |
| Chip design | AIG | 1 200 000 | 125.9 | 236.3 | 3.7 |
| Electronic circuits | 5 Components | 73 000 | 13.0 | 30.0 | 7.0 |
| | 7 Components | 14 000 | 17.0 | 42.0 | 9.3 |
| | 10 Components | 6 000 | 24.0 | 56.0 | 12.3 |
| SAT Solving | Small – VG | 69 596 | 1 323.96 | 38 791.52 | 33.9 |
| | Small – VCG | 69 596 | 8 304.22 | 45 402.51 | 5.4 |
| | Small – LCG | 69 596 | 9 628.12 | 45 402.05 | 4.75 |
| Combinatorial optimization | RB – Small | 50 000 | 241.0 | 9 016.1 | 36.9 |
| | RB – Large | 50 000 | 1 027.8 | 102 644.8 | 99.4 |
| | ER – Small | 50 000 | 249.9 | 9 463.5 | 37.3 |
| | ER – Large | 50 000 | 750.2 | 84 450.1 | 112.4 |
| | BA – Small | 50 000 | 249.9 | 991.6 | 3.9 |
| | BA – Large | 50 000 | 750.1 | 2 992.4 | 3.9 |
| Algorithmic reasoning-EASY | Topological Sorting | 1 020 000 | 16.0 / 16.0 /128.0 | 41.006 / 40.997 / 1121.008 | 2.563 / 2.562 / 8.758 |
| | MST | 1 020 000 | 16.0 / 16.0 /128.0 | 24.024 / 24.004 / 1147.972 | 1.5015/ 1.5003 / 8.9685 |
| | Bridges | 1 020 000 | 16.0 / 16.0 /128.0 | 20.754/ 20.794/ 277.08 | 1.2971 / 1.2996 / 2.1647 |
| | Steiner Trees | 1 020 000 | 16.0 / 16.0 /128.0 | 28.739 / 28.733 / 755.284 | 1.796 / 1.796 / 5.901 |
| | Max Clique | 1 020 000 | 16.0 / 16.0 /128.0 | 77.99 / 77.95 / 2698.5 | 4.875 / 4.872 / 21.081 |
| | Flow | 1 020 000 | 16.0 / 16.0 /64.0 | 75.76 / 75.84 / 530.96 | 4.734 / 4.740 / 8.296 |
| | Max Matching | 1 020 000 | 16.0 /16.0 / 128.0 | 38.890 / 38.897 / 746.453 | 2,431 / 2,431 / 5,834 |
| Algorithmic reasoning-MEDIUM | Topological Sorting | 1 000 000 | 16.0 / 16.0 /128.0 | 36.167 / 36.137 / 1097.05 | 2.260 / 2.259 / 8.571 |
| | MST | 1 020 000 | 16.0 / 16.0 /128.0 | 23.976 / 24.003 / 1140.312 | 1.498/ 1.50 / 8.909 |
| | Bridges | 1 020 000 | 16.0 / 16.0 /128.0 | 17.539/ 17.54/ 277.550 | 1.0962 / 1.0963 / 2.168 |
| | Steiner Trees | 1 020 000 | 16.0 / 16.0 /128.0 | 19,549 / 19.562 /742.301 | 1.222 / 1,223 / 5.799 |
| | Max Clique | 1 020 000 | 16.0 / 16.0 /128.0 | 108.0 / 108.0 / 2687.22 | 6.751 / 6.751 / 20.994 |
| | Flow | 1 020 000 | 16.0 / 16.0 /64.0 | 42.36 / 42.36 / 645.20 | 2.648 / 2.648 / 10.08 |
| | Max Matching | 1 020 000 | 16.0 / 16.0 /128.0 | 16.305 / 16.306 / 726,908 | 1,0189/ 1,020 /5,679 |
| Algorithmic reasoning-HARD | Topological Sorting | 1 020 000 | 16.0 / 16.0 /128.0 | 36.143 / 36.142 / 827.09 | 2.259 / 2.259 / 6.462 |
| | MST | 1 020 000 | 16.0 /16.0 /128.0 | 24.008/ 23.998/ 978.825 | 1,50 / 1.499 / 7.647 |
| | Bridges | 1 020 000 | 16.0 / 16.0 /128.0 | 17.539/ 17.54/ 277.08 | 1.096 / 1.096 / 2.165 |
| | Steiner Trees | 1 020 000 | 16.0 /16.0 /128.0 | 19.583 / 19.565 / 663.547 | 1.224 /1,223 / 5.184 |
| | Max Clique | 1 020 000 | 16.0 / 16.0 /128.0 | 108.01 / 107.99 / 1812.04 | 6.751 / 6.750 / 14.157 |
| | Flow | 1 020 000 | 16.0 / 16.0 /64.0 | 42.36 / 42.36 / 645.20 | 2.648 / 2.648 / 10.08 |
| | Max Matching | 1 020 000 | 16.0 /16.0 / 128.0 | 16,303 / 16,304 / 746,521 | 1,018/ 1,019 /5,832 |
| Weather forecasting | ERA5 64x32 | 93 544 | 4 610 | 59 667 | 12.9 |

## B  Dataset Details

### B.1  Social Sciences

**Related work**  A wide range of social network benchmarks are available in repositories such as SNAP (Leskovec & Sosic, 2016) and TUDATASETS (Morris et al., 2020), yet these datasets diverge substantially from real-world social networks. Benchmarks such as FACEBOOK, GITHUB, TWITCH, LASTFM, DEEZER, and REDDIT (Rozemberczki et al., 2021; Hamilton et al., 2017) illustrate the main issues: (1) interactions are collapsed into static graphs, eliminating temporal dynamics; (2) prediction targets are categorical (e.g., user type, demographic attribute) rather than continuous behavioral outcomes; (3) topologies are simplified to undirected friendships or follows, omitting richer relations such as replies, reposts, or pull requests; (4) node features are weak or outdated, often sparse indicators or embeddings from GloVe/Word2Vec; and (5) preprocessing alters network structure, for example by removing large communities in REDDIT, thereby biasing degree distributions and interaction patterns.

The TUDATASET collection shares similar limitations. Their REDDIT variants represent discussion threads as featureless graphs with reply edges and categorical thread-level labels, while COLLAB and IMDB construct ego-networks labeled by field or genre. These datasets are likewise static, categorical, topologically restricted, feature-poor, and further constrained by artificial graph boundaries that remove long-range dependencies.

In contrast, our datasets are designed to support more realistic forecasting. We provide temporally consistent splits, continuous engagement-based targets, and interaction topologies that preserve multiple directed relation types. Node features are derived from user-generated text using modern language-model embeddings, and networks are released without *arbitrary* pruning, maintaining original degree distributions and structural diversity.

**Evaluation of the learning task**  We assess model performance using two metrics: the coefficient of determination ($R^2$) and the Spearman correlation ($\rho$). Given a set of evaluation nodes $U \subset V$, reference engagement kind $\kappa$ and prediction interval $\tau_{1,2}$, the former is defined as

$$R^2_{\kappa,\tau_{1,2}} := 1 - \frac{\sum_{u \in U}(y^{\kappa,(u)}_{\tau_{1,2}} - \hat{y}^{\kappa,(u)})^2}{\sum_{u \in U}(y^{\kappa,(u)}_{\tau_{1,2}} - \bar{y}^{\kappa}_{\tau_{1,2}})^2},$$

where $\hat{y}^{\kappa,(u)}$ is the model's prediction for engagement $\kappa$ and user $u$ and $\bar{y}^{\kappa}_{\tau_{1,2}}$ is the mean target value over $U$. This standard regression metric indicates the overall variance captured by the model, enabling the assessment of its prediction accuracy compared to that of a trivial predictor. The Spearman correlation metric is defined, instead, as:

$$\sigma_{\kappa,\tau_{1,2}} := 1 - \frac{6 \cdot \sum_{u \in U}\left(R[y^{\kappa,(u)}_{\tau_{1,2}}] - R[\hat{y}^{\kappa,(u)}]\right)^2}{n_U(n_U^2 - 1)},$$

where $n_U = |U|$ and expression $\left(R[y^{\kappa,(u)}_{\tau_{1,2}}] - R[\hat{y}^{\kappa,(u)}]\right)$ evaluates the difference in the ranking of user $u \in U$ within either targets ($y^{\kappa}_{\tau_{1,2}}$) or model's predictions ($\hat{y}^{\kappa}$). This metric is particularly relevant in our setting, as it measures how well predictions preserve the relative ordering of users by future engagement. This is a key criterion when ranking influence in heavy-tailed social networks and is not necessarily captured by accuracy metrics such as $R^2$.

**Details on the dataset**  We construct our benchmark using data from the BLUESKY social network,[2] leveraging the publicly curated dataset described in (Failla & Rossetti, 2024). We refer the reader to their paper for full details on the data collection and curation process. In the following, we summarize the components relevant to our benchmark.

**Graph structures**  The dataset contains three different graph structures that capture network-wide *interactions* based on quotes, replies, and reposts: a *directed* edge from node $u$ to node $v$ indicates that, respectively, $u$ quoted, replied to, or reposted a $v$'s post. Note that these interactions are originally timestamped; aiming at preventing data leakage, we employ this information to construct graph structures

---

[2]https://bsky.app/

that are temporally consistent with the proposed time-based splits, i.e., made up only of interactions in $\tau_{0,1} = (t_0, t_1]$ as defined in Section 3.1. We only consider one single edge when multiple interactions are present in $\tau_{0,1}$ for the same user pair. In future iterations of this dataset, we will consider endowing edges with additional features conveying the amount and time distribution of such interactions.

**Node features**    Each node in the above graph structures corresponds to a user, which we describe with an aggregated representation of the content of their recent posts. Specifically, for each user, we process the text obtained by concatenating the content of each of their posts at the monthly granularity. We then employ the `sentence-transformers/all-MiniLM-L6-v2` language model [3] to embed these and obtain aggregated textual representations, which we subsequently aggregate (by averaging) over the time interval $(t_0, t_1]$; see above Section 3.1.

**Node targets**    are calculated as the *median* number of engagements obtained by the user's posts, in particular likes, replies, and reposts. These are considered as separate prediction targets and are measured in the time interval $(t_1, t_2]$. We apply a logarithmic transformation to reduce the skewness of these prediction targets.

**Splitting procedure**    The splitting strategy ensures that models are trained exclusively on past information and evaluated on later interactions. The posts considered to obtain node features and targets span a time interval from February 17th, 2023 ($t_{\text{start}}$) to March 18th, 2024 ($t_{\text{end}}$). On top of these, we consider three key time points:

- $t_A$: December 11th, 2023;
- $t_B$: January 22nd, 2024;
- $t_C$: February 20th, 2024.

These are obtained in a way that the proportion of posts in $\tau_{\text{start},A}, \tau_{A,B}, \tau_{B,C}, \tau_{C,D}$ amounts to, resp., $55\%, 15\%, 15\%, 15\%$. The dataset is then accordingly split in training, validation, and test splits as follows:

- Training: $t_0 = t_{\text{start}}, t_1 = t_A, t_2 = t_B$;
- Validation: $t_0 = t_{\text{start}}, t_1 = t_B, t_2 = t_C$;
- Test: $t_0 = t_{\text{start}}, t_1 = t_C, t_2 = t_{\text{end}}$.

While ensuring the model is not trained on post contents and network interactions from the future, by setting $t_0 = t_{\text{start}}$ in all splits this partitioning strategy also reflects a realistic scenario where a social network "grows in time", in the sense that user representations evolve as they generate new content and their connections expand as they interact with more users.

## B.2 HARDWARE DESIGN

### B.2.1 CHIP DESIGN

Chip design is among the most complex engineering challenges today, with significant implications for the global economy and supply chains (Ou et al., 2024). Fabricating a competitive system-on-chip requires arranging billions of transistors under strict performance, power, and area (PPA) constraints. This process is orchestrated by electronic design automation (EDA) tools, which automate key stages from high-level synthesis to physical layout. Yet, as design scale and complexity continue to grow, bottlenecks in design space exploration and functional verification are becoming increasingly severe.

Logic synthesis is a fundamental step within the EDA flow, translating a behavioral specification into a structural implementation of interconnected logic gates (Rai et al., 2021). The resulting Boolean circuit provides the core representation of a system's logic. Optimizing this circuit, typically by minimizing its size (area), is a well-known NP-hard problem (Ilango et al., 2020). For decades, progress has relied on carefully engineered heuristics and combinatorial optimization techniques embedded in EDA tools. However, these classical approaches are reaching diminishing returns, and their computational costs scale poorly with modern circuit complexity (Tsaras et al., 2025).

---

[3] `https://huggingface.co/sentence-transformers/all-MiniLM-L6-v2`

Table 6: Statistics of the synthetic AIG dataset.

| #Graphs | #Inputs | #Outputs | Mean #Nodes | Median #Nodes | Max. #Nodes |
|---------|---------|----------|-------------|---------------|-------------|
| 1.2M | 6-8 | 1-2 | 125.9 | 104.0 | 335.0 |

Recently, machine learning has begun to influence key stages of chip design, including macro placement (Mirhoseini et al., 2021) and floor planning (Mallappa et al., 2024). In this work, we propose focusing such efforts on logic synthesis, framing it as a conditional graph generation problem: generating a circuit that correctly implements a given Boolean function while minimizing the number of gates. The goal is thus to produce circuits that satisfy a prescribed truth table exactly while optimizing structural efficiency.

Formally, let $f: \{0,1\}^n \to \{0,1\}^m$ be a Boolean function with $n$ inputs and $m$ outputs. Its behavior is fully specified by a truth table $\boldsymbol{T}^f \in \{0,1\}^{m \times 2^n}$, where each column $\boldsymbol{T}^f_{\boldsymbol{x}}$ corresponds to an input $\boldsymbol{x} \in \{0,1\}^n$ and satisfies $f(\boldsymbol{x}) = \boldsymbol{T}^f_{\boldsymbol{x}}$. A logic circuit $C := (V, E)$ is represented as a *directed acyclic graph* (DAG) of in-degree at most two. The node set is partitioned into input nodes $V^{\text{in}} := \{v^{\text{in}}_1, \ldots, v^{\text{in}}_n\}$, gate nodes $V^{\text{gate}} := \{v^{\text{gate}}_1, \ldots, v^{\text{gate}}_k\}$, each computing a Boolean function of arity at most two (e.g., AND), and output nodes $V^{\text{out}} := \{v^{\text{out}}_1, \ldots, v^{\text{out}}_m\}$, each of in-degree one. The edge set $E \subseteq V \times V \times \{0,1\}$ specifies connections, where a label of 1 denotes signal inversion (NOT) and 0 denotes direct transmission. For an input $\boldsymbol{x} \in \{0,1\}^n$, values are assigned to input nodes, propagated through gate nodes (applying inversion where specified), and collected at output nodes to yield the circuit output $C(\boldsymbol{x}) \in \{0,1\}^m$. Because the combination of AND and NOT is functionally complete, any Boolean function can be represented using only these two operations. We therefore adopt the standard representation of *and-inverter graphs* (AIGs), where circuits are DAGs of AND gates connected by edges that may invert their signals (Mishchenko et al., 2005). AIGs are widely used in industrial logic synthesis and provide a compact, canonical representation of circuits.

With this benchmark, we introduce a new family of graphs paired with a challenging optimization objective: models must generate valid tree-like DAGs that not only satisfy a target truth table but also minimize circuit size. This setting provides a rigorous testbed for evaluating the capabilities of generative graph methods under realistic and computationally challenging constraints.

**Related work**   AlphaChip (Mirhoseini et al., 2021) kickstarted efforts to bring the latest machine learning advancements into a relatively conservative area of chip design, applying RL to the macro placement task and outperforming human engineers on specific metrics, as well as processing designs for newer generations of TPUs (Goldie et al., 2024). In logic synthesis, a considerable amount of work has tackled discriminative objectives (Zheng et al., 2025) rather than generative ones. One of the main benchmarks in the field is the contest organized by the *International Workshop on Logic & Synthesis* (IWLS) (Mishchenko & Miyasaka, 2023) where participants are asked to optimize 100 pre-selected circuits. Historically, combinatorial optimization methods prevailed due to the lack of training splits and the limited amount of available data. However, recent machine learning methods (Li et al., 2025b) started showing promising results by training on large amounts of synthetic data simulated with standard EDA tools.

**Details on the dataset**   For this benchmark, we require paired datasets of truth tables and their corresponding AIGs to train generative models. To construct such a dataset, we first generated 1,200,000 random truth tables in the PLA format (Wille et al., 2008), each with 6–8 inputs and 1 or 2 outputs. These truth tables were then compiled into optimized AIGs in the AIGER format (Biere et al., 2011) using ABC. During this step, we applied a sequence of standard optimization commands (`strash`, `resyn2`, `resyn2`, `dc2`, `resyn2rs`, `resyn2`). We subsequently converted the AIGs into DOT format (Gansner et al., 2006) via the AIGER library, from which they can be conveniently transformed into PYTORCH GEOMETRIC graphs. Alongside these graph representations, we store the truth tables explicitly as matrices of size $m \times 2^n$, where $n$ is the number of inputs and $m$ the number of outputs. We split the dataset into train, validation, and test sets using an 80/10/10 split. Details concerning the dataset are presented in Table 6.

**Evaluation**   We report results using four standard ABC variants:

- STRASH converts the circuit into an And-Inverter Graph representation using structural hashing. This serves as the basis for subsequent optimizations.

- RESYN is a lightweight synthesis script that alternates between balancing and rewriting.
- COMPRESS2 applies balancing, rewriting and refactoring, together with zero-cost replacements, to restructure the network and enable further simplifications.
- RESYN2RS is a heavyweight optimization script that performs multiple rounds of balancing, rewriting, refactoring, and resubstitution, progressively increasing the cut sizes ($K = 6$ to $12$).

As you may notice, the methods are ordered by their level of complexity, with STRASH being the simplest and lightest, and RESYN2RS the most complex and expensive.

**Results**    Our results using four different ABC variants are reported in Table 7. We report the score defined in Equation (1). The methods rank consistently with their complexity across all settings, with the simple STRASH yielding the worst results and the advanced RESYN2RS achieving the best performance.

Table 7: **Chip Design.** Results of the different ABC methods on the AIG dataset. Higher is better.

| Method | Number of (inputs, outputs) | | | | | | |
| --- | --- | --- | --- | --- | --- | --- | --- |
| | (6,1) | (6,2) | (7,1) | (7,2) | (8,1) | (8,2) | Overall |
| STRASH | 73.90 | 74.23 | 76.55 | 75.71 | 77.63 | 76.52 | 75.76 |
| RESYN | 86.79 | 88.16 | 89.28 | 89.05 | 90.45 | 89.78 | 88.92 |
| COMPRESS2 | 92.53 | 93.57 | 93.18 | 92.79 | 93.28 | 92.49 | 92.97 |
| RESYN2RS | 93.30 | 95.24 | 95.15 | 95.64 | 96.24 | 96.11 | 95.28 |

### B.2.2 ELECTRONIC CIRCUITS

Analog circuit topology design is a representative open challenge in EDA. Even with modest component counts, the number of feasible topologies grows combinatorially, quickly reaching hundreds of thousands of configurations. Despite the continued introduction of new converter families, their development remains guided mainly by human intuition. Learning-based approaches offer a promising alternative by serving as surrogate models that predict circuit performance, accelerating evaluation and guiding search in vast design spaces. Power converters are naturally expressed as graphs, making them a compelling testbed for graph learning. The task stresses model robustness to structural sensitivity (small topological changes can induce large performance shifts) and class imbalance (high-performing circuits are rare), providing a challenging and practically relevant benchmark.

**Related work**    Learning for circuit design has been explored for physical implementation (Lu et al., 2020) and parameter optimization (Wang et al., 2020), typically assuming fixed topologies. Topology exploration, in contrast, has relied on heuristic search, including evolutionary and genetic algorithms (McConaghy et al., 2011), as well as tree-based strategies (Fan et al., 2021; Zhao & Zhang, 2020a). Our benchmark differs in that it casts performance prediction as a supervised graph learning task, utilizing standardized splits and metrics, which enables systematic comparison across architectures.

**Description of the learning task**    Each circuit is represented as a graph $G = (V, E)$. The node set $V$ contains device components—capacitors ($C$), inductors ($L$), and two phase-specific switches ($S_a, S_b$)—and three external terminal ports: input ($V_{\text{in}}$), output ($V_{\text{out}}$), and ground (GND). Each component node $v \in V$ has a type $\tau(v)$ and (optional) parameter vector $d_v$ and connects to other devices or ports via exactly two edges; each terminal node has a single outgoing edge. Edges $E$ denote electrical interconnections between ports.

Given a graph $G$, the prediction task is to estimate two continuous performance metrics: (i) the *voltage conversion ratio* (output-to-input voltage) and (ii) the *power conversion efficiency* (fraction of input power delivered to the load). Performance is evaluated via the *relative squared error* (reported separately for each target)

$$\text{RSE}(y, \hat{y}) := \frac{\sum_{i=1}^{N}(y_i - \hat{y}_i)^2}{\sum_{i=1}^{N}(y_i - \bar{y})^2},$$

where $y_i$ is the ground-truth value from high-fidelity simulation, $\hat{y}_i$ the model prediction, $\bar{y}$ the mean of $\{y_i\}_{i=1}^{N}$, and $N$ the number of evaluation samples.

**Details on the dataset** Following the setting proposed by Fan et al. (2024b), we used datasets at three complexity levels, generated from random valid topologies with *5*, *7*, and *10* components. Topologies are sampled uniformly at random under connectivity constraints; graph isomorphisms are removed to ensure uniqueness. Each instance is simulated with NGSPICE (Nenzi & Vogt, 2011) to obtain ground-truth voltage conversion ratio and efficiency; instances flagged as invalid during simulation are discarded. Unless otherwise specified, component and operating parameters are fixed as follows: capacitors $10\,\mu\text{F}$, inductors $100\,\mu\text{H}$, input voltage $100\,\text{V}$, switching frequency $1\,\text{MHz}$, an input resistor of $0\,1\,\Omega$ at $V_{\text{in}}$, and a $100\,\Omega$ load with a $10\,\mu\text{F}$ output capacitor at $V_{\text{out}}$. The resulting datasets contain approximately 73k, 14k, and 6k instances for the 5-, 7-, and 10-component settings, respectively. Graphs typically comprise more than 13 internal nodes (plus three terminals) for 5-component circuits, and over 17 and 24 internal nodes for the 7- and 10-component sets, respectively.

**Results** We conduct experiments on datasets containing 7- and 10-component circuits, extending beyond the 5-component dataset. In each experiment, a model is trained from scratch and evaluated on a dataset containing circuits of the same size. Table 8 presents the RSE for efficiency and output voltage prediction, comparing GT with baseline models (GCN, GAT, and GIN) as circuit sizes increase.

Table 8: Performance comparison of baseline models on circuit datasets with 5, 7, and 10 components (RSE).

| Task | Method | Size | | |
|---|---|---|---|---|
| | | 5 comp | 7 comp | 10 comp |
| Efficiency | GT | $0.07\pm 0.03$ | $0.18\pm 0.06$ | $0.29\pm 0.03$ |
| | GIN | $0.09\pm 0.06$ | $0.21\pm 0.04$ | $0.44\pm 0.13$ |
| | GAT | $0.11\pm 0.05$ | $0.22\pm 0.05$ | $0.38\pm 0.05$ |
| | GCN | $0.13\pm 0.07$ | $0.30\pm 0.04$ | $0.53\pm 0.06$ |
| Voltage | GT | $0.12\pm 0.02$ | $0.31\pm 0.05$ | $0.39\pm 0.05$ |
| | GIN | $0.14\pm 0.05$ | $0.35\pm 0.03$ | $0.54\pm 0.12$ |
| | GAT | $0.17\pm 0.13$ | $0.34\pm 0.12$ | $0.48\pm 0.02$ |
| | GCN | $0.18\pm 0.07$ | $0.46\pm 0.08$ | $0.61\pm 0.07$ |

## B.3 REASONING AND OPTIMIZATION

### B.3.1 SAT SOLVING: ALGORITHM SELECTION AND PERFORMANCE PREDICTION

The *Boolean satisfiability problem* (SAT) is a central and longstanding NP-complete problem (Karp, 1972; Biere et al., 2021). It asks whether there exists a truth assignment that satisfies a Boolean formula $\varphi$ over variables $v_1, \ldots, v_r$, typically expressed in *conjunctive normal form* (CNF):

$$\varphi = \bigwedge_{i=1}^{n} c_i, \quad \text{where each } c_i = \bigvee_{j=1}^{m_i} l_{i,j}.$$

Here each $l_j$ is a literal, i.e., either a variable $v_p$ or its negation $\neg v_p$. The task is to decide whether there exists an assignment $a$ such that $\varphi(a) = \text{true}$.

SAT is of significant theoretical importance as one of the first problems proven NP-complete (Cook, 1971), and it underpins numerous practical applications in hardware and software verification (Biere et al., 2009), automated planning (Kautz & Selman, 1996), and operations research (Gomes et al., 2008). Its significance has led to the development of a wide range of solvers, systematically benchmarked in annual SAT competitions (e.g., (Froleyks et al., 2021; Balyo et al., 2017; Heule et al., 2019)).

A key property of modern SAT solvers is *performance complementarity*—no single solver dominates across all instances. This motivates the study of *algorithm selection* (Rice, 1976; Xu et al., 2008), where the aim is to choose the most effective solver for a given instance. *Empirical performance prediction* is a closely related problem, which seeks to forecast the computation time (or another performance metric) of a solver on a given instance. Such predictions can support algorithm selection, configuration, scheduling, and explainability (Hutter et al., 2014a).

State-of-the-art methods for algorithm selection and performance prediction rely on *hand-crafted features* extracted from SAT instances. The most widely adopted feature set is that of the SATZILLA family of algorithm selectors (Nudelman et al., 2004; Hutter et al., 2014a; Shavit & Hoos, 2024). These features include basic structural properties (e.g., number of clauses and variables), graph-based descriptors derived from instance structure, and probing features obtained by briefly running a solver to gather computation time statistics. Machine learning models—most often tabular models—are then trained on these features for prediction or selection tasks.

**Motivation for inclusion in GraphBench**   SAT instances can be naturally represented as graphs, reflecting their permutation-invariant structure that arises from the commutativity and associativity of logical conjunction and disjunction. Indeed, many of the hand-crafted SATZILLA features are derived from such graph representations, including the

- *variable-clause graph*: a bipartite, undirected graph with a node for each variable $v$ and each clause $c$, where an edge connects a variable to a clause if and only if the variable appears in that clause.
- *Variable graph*: an undirected graph with one node per variable, where two variables $v_i$ and $v_j$ are connected if they co-occur in at least one clause.
- *Clause graph*: an undirected graph with one node per clause, where two clauses $c_i$ and $c_j$ are connected if they share at least one variable (possibly negated).

SATZILLA extracts statistical properties of these graphs—such as mean degree, coefficient of variation, clustering coefficient, and graph diameter—and uses them as features for downstream learning tasks.

Building on this success, several studies have explored the use of GNNs or MPNNs to predict the satisfiability of a formula, e.g., (Selsam et al., 2019). Some of this work employs an additional representation, the *literal–clause graph*, a bipartite graph with one node per literal (a variable or its negation) and one node per clause, with edges linking literals to the clauses they appear in. However, many such approaches are *unsound*—they can produce incorrect results—or cannot guarantee valid satisfying assignments or proofs of unsatisfiability, limiting their practical usefulness (Selsam et al., 2019; Li et al., 2024c).

Other research has integrated MPNNs into existing SAT solvers to replace or guide solver heuristics (Wang et al., 2024; Tönshoff & Grohe, 2025). These approaches retain the soundness and completeness guarantees of classical solvers while potentially accelerating the solving process. Nonetheless, benchmarking novel MPNN architectures in this setting is challenging, as it requires extensive pretraining and large-scale evaluations.

A complementary line of work applies MPNNs to algorithm selection in SAT (Zhang et al., 2024; Shavit, 2023), predicting the most effective solver for a given instance directly from graph-based representations.

In GRAPHBENCH, we introduce, to the best of our knowledge, the largest algorithm-selection benchmark for SAT solvers, covering eleven solvers and more than 100 000 instances. Our benchmark supports multiple graph representations of SAT formulae and also provides the SATZILLA 2024 feature set (Shavit & Hoos, 2024), enabling their combination with MPNNs. Unlike existing GNN benchmarks for SAT (Li et al., 2024c), which primarily focus on satisfiability prediction or assignment generation, our benchmark is designed for practical downstream tasks in SAT solving—specifically, algorithm selection and performance prediction.

**Related work**   As the usage of MPNNs for the task of algorithm selection is a relatively new research direction, no benchmarks exist for the task. Previously, Shavit (2023) used a dataset of 3 000 synthetically generated SAT instances, while Zhang et al. (2024) used a proprietary dataset as well as the 2022 Anniversary track of the SAT Competition, while removing large instances.

Outside of the graph domain, the ASLIB benchmark suite is used to benchmark algorithm selection methods (Bischl et al., 2016). ASLIB contains multiple algorithm selection datasets from various domains, including SAT and TSP, among others. Each dataset contains features extracted from the instances (such as SATZILLA) as well as the computation time of several algorithms on the instances. We note that ASLIB offers SAT scenarios based on a subset of SAT competitions, containing the instances and solvers from specific years. In contrast, our dataset comprises all available SAT competition instances, as well as numerous instances from other sources.

**Description of the learning task** We introduce two learning tasks: *performance prediction* (a regression problem) and *algorithm selection* (a multi-class classification problem). While algorithm selection can be reduced to performance prediction by choosing the solver with the lowest predicted computation time, we treat them as distinct problems. For algorithm selection, we adopt the loss function proposed by (Zhang et al., 2024), which penalizes the predicted probabilities in proportion to the solver's computation time. Intuitively, a solver that only slightly underperforms on an instance incurs a small penalty, while poor selections are penalized more heavily:

$$\frac{1}{N} \sum_{i=1}^{N} \left( \sum_{k=1}^{K} p_i^k \cdot t_i^k - t_i^* \right)^2,$$

where $p_i^k$ is the predicted probability of selecting solver $k$ for instance $i$, $t_i^k$ is the computation time of solver $k$ on instance $i$, and $t_i^* = \min_k t_i^k$ is the computation time of the virtual best solver (VBS).

For the SAT benchmarks, each instance is provided with the four graph-based representations described above. In all cases, SATZILLA features can be integrated as node attributes to enrich the graph representation.

The performance of a solver $s$ on an instance set $\Pi = \{\pi_1, \pi_2, \ldots, \pi_n\}$ is measured using the widely adopted *penalized average computation time* ($\text{PAR}_k$) metric, where $k$ denotes the penalty factor for unsolved instances under a cutoff time $c$. The $\text{PAR}_k(s, \Pi)$ score is defined as the mean computation time of $s$ across $\Pi$, where each unsolved instance contributes a computation time of $k \cdot c$. Lower $\text{PAR}_k$ values indicate better performance. For algorithm selectors, both feature extraction and model inference time are included in the total solving time, implying that GNN inference must be efficient to be practically helpful.

For the performance prediction task, the model output is the base-10 logarithm of the $\text{PAR}_{10}$ score of solver $s$:

$$\log_{10}(\text{PAR}_{10}(s, \Pi)),$$

a transformation shown to improve predictive accuracy in prior work (Hutter et al., 2014a). Model performance is evaluated using the root mean squared error (RMSE) between predicted and actual log-scaled computation times.

For algorithm selection, we use the *closed gap* (CG) metric, which quantifies how close a selector comes to the performance of the VBS relative to the single best solver (SBS). Formally,

$$\text{CG}(\Pi) = \frac{\text{PAR}_{10}(\text{SBS}, \Pi) - \text{PAR}_{10}(s, \Pi)}{\text{PAR}_{10}(\text{SBS}, \Pi) - \text{PAR}_{10}(\text{VBS}, \Pi)}.$$

A higher CG score indicates stronger selection performance, with $\text{CG} = 1$ corresponding to VBS performance and $\text{CG} = 0$ to SBS performance.

**Details on the dataset** For this benchmark, we created the largest algorithm selection scenario for SAT solvers available to date. We collected all SAT Competition instances via GBD (Iser & Jabs, 2024) and added all instances from the AClib benchmark for algorithm configuration (Hutter et al., 2014b). Together, these sources include SAT formulae from diverse real-world applications as well as synthetically generated instances. To further enrich the dataset, we generated additional formulae using three well-established SAT instance generators:

- *FuzzSAT* (Brummayer et al., 2010) produces CNF fuzzing instances originally designed to identify issues in digital circuits. It generates random Boolean circuits as DAGs and converts them into CNF via the Tseitin transformation (Tseitin, 1983). To construct the DAG, we used $v \in [50, 150]$ input variables; operands were randomly paired and connected by randomly chosen operator nodes until each input was referenced at least $t \in [1,5]$ times. Finally, random clauses of length $s \in [3, 8]$ were added to increase complexity.

- *Cryptography* (Nejati et al., 2017; Alamgir et al., 2024) generates SAT instances encoding preimage attacks on MD4, SHA-1, and SHA-256. Each encoding applies multiple *rounds* of operations; higher numbers of rounds yield harder instances. We set the number of rounds to $d \in [16, 30]$, which produces formulae challenging for modern solvers but not unsolvable. Additionally, we fixed $i \in [0,384]$ of the 512 input bits; fixing more bits results in easier instances.

Table 9: Overview of the instances for the SAT solving benchmark.

| Source | # of Instances | # of Variables | | | | # of Clauses | | | |
|---|---|---|---|---|---|---|---|---|---|
| | | min | max | avg | median | min | max | avg | median |
| SAT Competition | 31 024 | 3 | 25 870 369 | 69 190.77 | 13 168.50 | 7 | 871 935 536 | 1 022 025.50 | 118 526.00 |
| Community Attachment | 29 994 | 922 | 2 956 | 1 931.05 | 1 928.00 | 3 566 | 13 413 | 8 088.71 | 8 041.00 |
| AClib | 33 955 | 6 | 248 738 | 3 319.65 | 1 270.00 | 24 | 103 670 100 | 136 471.03 | 10 000.00 |
| FuzzSAT | 8 020 | 2 | 944 685 | 10 217.75 | 1 820.00 | 1 | 4 094 516 | 45 638.30 | 8 074.50 |
| Cryptography | 4 873 | 6 | 12 415 | 4 264.89 | 3 225.00 | 24 | 185 006 | 55 309.63 | 45 316.00 |
| **Total** | 107 866 | 2 | 25 870 369 | 22 434.71 | 1 879.00 | 1 | 871 935 536 | 345 051.72 | 10 177.50 |

- *Community attachment* (Giráldez-Cru & Levy, 2016) produces pseudo-random SAT formulae with community structure, a property often observed in real-world SAT encodings (e.g., hardware and software verification tasks (Ansótegui et al., 2012)). The generator can create relatively small but difficult instances. We used $n \in [1\,000, 3\,000]$ variables, $c \in [5, 100]$ communities, a clause-to-variable ratio of $[3\,7, 4\,5]$, and modularity factor $q \in [0\,3, 0\,9]$. Higher modularity values enforce stronger community structure.

For each generator, parameter ranges were chosen to yield challenging but still solvable instances. To ensure this, we excluded ranges of values yielding no instances solvable by KISSAT, the winner of the 2024 SAT Competition, between 1 and 5 000 seconds. After generation, we applied the SATELITE preprocessor to remove redundant clauses and variables. A summary of all generated instances is provided in Table 9.

We pre-selected solvers for the benchmark by constructing a portfolio of $m$ complementary state-of-the-art SAT solvers, based on the results of the 2024 SAT Competition. Portfolios were built using beam search to identify the set of $m$ solvers achieving the best virtual best solver (VBS) performance. To determine $m$, we evaluated portfolios of all possible sizes and plotted portfolio size against achieved VBS. We then identified the inflection point (the "knee") of this curve using the Kneedle algorithm (Satopaa et al., 2011) and adopted the corresponding portfolio. The ASF library (Shavit & Hoos, in progress) was used to perform the selection procedure. The same procedure was repeated on the SAT Competition 2023.

In total, our dataset includes the following solvers. From the 2024 SAT Competition: Kissat (Biere et al., 2024), BreadIdKissat (Bogaerts et al., 2024), AMSAT (Li et al., 2024b), and KissatMABDC (Liu et al., 2024). From the 2023 SAT Competition: SBVA CadiCal (Haberlandt & Green, 2023), Reencode Kissat (Reeves & Bryant, 2023), Minisat XOR, KissatMAB Prop PR (Gao, 2023), hKissatInc (Tchinda & Djamegni, 2023), CadiCal vivinst (Biere et al., 2023), and BreakID Kissat (Bogaerts et al., 2023). Solver computation times were measured in terms of CPU time using `runsolver` (Roussel, 2011) and the generic wrapper tools (Eggensperger et al., 2019), ensuring accurate and consistent performance measurement.

We evaluate the performance of the baselines on all SAT-solving-related benchmarks, as described in Appendix B.3.1. The following paragraphs describe the evaluation pipeline, additional (non-graph-based) baselines, and the results.

**Additional baselines**  For the performance prediction task, we compare the MPNNs against traditional baselines, which use the SATZILLA features and a (tabular) machine learning model. We use random forest (Breiman, 2001) and XGBoost (Chen & Guestrin, 2016), which are commonly used as empirical performance models (EPM) (see, e.g., Bansal et al. (2022); Eggensperger et al. (2015); Hutter et al. (2014c)). We use the SATZILLA 2024 features as input and predict the log10-scaled computation time. These baselines are implemented using the ASF framework (Shavit & Hoos, in progress). For algorithm selection, we provide several well-known baselines. First, we use pairwise regression (Kerschke et al., 2018) (PR; predicts performance differences between each pair of solvers, then picks the solver with the best sum), pairwise classification (Xu et al., 2012) (PC; predicts the better solver in each pair, then uses majority vote, winner of the ICON challenge on algorithm selection (Lindauer et al., 2019)) as well as a simple multi-class classification (Kotthoff, 2013) (MC; directly predicts the best solver). Similar to the EPM baselines, these baselines use the SATZILLA 2024 features with a random forest.

We furthermore considered AutoFolio (Lindauer et al., 2015) as a baseline. However, the smallest dataset we introduce contains more than 60 000 samples, whereas typical algorithm selection datasets (e.g., those in ASlib (Bischl et al., 2016)) have only hundreds to thousands of instances. As a result, AutoFolio failed to provide meaningful results and consistently predicted the single best solver. Additionally, the winner of the 2017 Open Algorithm Selection Challenge (Lindauer et al., 2019), ASAP (Gonard et al., 2017), lacks

support for setting the pseudo-random seed and only allows cross-validation evaluation; therefore, we excluded it.

**Evaluation** All SAT solving tasks in the benchmark are graph-level. For the graph transformer baselines, we employ an encoder-processor-decoder pipeline, as described in Appendix D, with no edge or node encoding. While we considered GT for the tasks, we encountered high times for the calculation of the positional embedding (more than a CPU day) and therefore excluded it for the evaluation. Furthermore, when training on medium-sized and large graphs, we encountered out-of-memory errors even on H100 with mixed precision training. Consequently, we only consider small formulae for GNN. Similarly, we observed out-of-memory with the clause graphs, and therefore excluded them from the evaluation.

We note that traditionally, algorithm selection datasets are evaluated using cross-validation, primarily due to limited data availability. In contrast, our new dataset contains tens of thousands of formulas. Since it is also intended for deep learning approaches, which are computationally expensive and thus impractical to evaluate using 10-fold cross-validation, we instead provide a fixed train–validation–test split and report results across multiple random seeds.

Table 10: **SAT Solving.** RMSE ($\log_{10}$ time) for performance prediction on SMALL SAT instances. We report results on three different graph representations (VG, VCG, LCG) and using the hand-crafted SATzilla features. Lower is better.

| Solver | Method | VG | VCG | LCG | SATZILLA |
|---|---|---|---|---|---|
| KISSAT | GIN | $1.36 \pm 0.15$ | $1.15 \pm 0.03$ | $1.49 \pm 0.02$ | – |
| | RF | – | – | – | $0.61 \pm 0.00$ |
| | XGB | – | – | – | $0.63 \pm 0.00$ |
| BREAKIDKISSAT | GIN | $1.29 \pm 0.03$ | $1.33 \pm 0.15$ | $1.52 \pm 0.01$ | – |
| | RF | – | – | – | $0.65 \pm 0.00$ |
| | XGB | – | – | – | $0.68 \pm 0.01$ |
| KISSATMABDC | GIN | $1.27 \pm 0.00$ | $1.43 \pm 0.24$ | $1.60 \pm 0.01$ | – |
| | RF | – | – | – | $0.70 \pm 0.00$ |
| | XGB | – | – | – | $0.74 \pm 0.00$ |
| AMSAT | GIN | $1.43 \pm 0.01$ | $1.45 \pm 0.09$ | $1.54 \pm 0.00$ | – |
| | RF | – | – | – | $0.68 \pm 0.00$ |
| | XGB | – | – | – | $0.72 \pm 0.01$ |

Table 11: **SAT Solving.** RMSE ($\log_{10}$ time) for performance prediction on MEDIUM and LARGE SAT instances. Lower is better.

(a) Medium-size SAT instances

| Solver | Method | SATZILLA |
|---|---|---|
| KISSAT | RF | $0.59 \pm 0.00$ |
| | XGB | $0.62 \pm 0.00$ |
| BREAKIDKISSAT | RF | $0.64 \pm 0.00$ |
| | XGB | $0.67 \pm 0.00$ |
| KISSATMABDC | RF | $0.68 \pm 0.00$ |
| | XGB | $0.72 \pm 0.00$ |
| AMSAT | RF | $0.64 \pm 0.00$ |
| | XGB | $0.67 \pm 0.00$ |

(b) Large SAT instances

| Solver | Method | SATZILLA |
|---|---|---|
| KISSAT | RF | $0.58 \pm 0.00$ |
| | XGB | $0.63 \pm 0.00$ |
| BREAKIDKISSAT | RF | $0.62 \pm 0.00$ |
| | XGB | $0.67 \pm 0.00$ |
| KISSATMABDC | RF | $0.66 \pm 0.00$ |
| | XGB | $0.70 \pm 0.00$ |
| AMSAT | RF | $0.62 \pm 0.00$ |
| | XGB | $0.66 \pm 0.00$ |

**Results** We start by presenting the results for the performance prediction task. We present the RMSE of the log-scaled computation time for the solvers from SAT Competition 2024 (KISSAT, BREAKIDKISSAT, KISSATMABDC, AMSAT) for the small formulae in table Table 10. The table shows that traditional empirical performance predictors (random forest and XGBoost) that use hand-crafted features outperform GNN by a high margin. Furthermore, we observe significant differences between the various graph types. While usage of LCG is common as an input for GNNs in the context of SAT solving (Selsam et al., 2019; Tönshoff & Grohe, 2025), the smaller VCG and VG perform better, which shows that the choice of input graph can have a high influence on the results. We furthermore present the results of random forest and

XGBoost on medium-size and large formulae in Tables 11a and 11b, such that they can be compared against graph machine learning solutions in the future. Overall, we observe that the random forest is the best-performing EPM for SAT solvers.

For the algorithm selection task, we report the closed gap metric of different selectors in Table 12. The VG representation allows for better performance than Both VCG and LCG, although using GIN with all three graph representation approaches, underperform compared to pairwise regression with hand-crafted features. Furthermore, VG also outperforms multi-class classification and achieves a slightly smaller gap closure than pairwise classification. In contrast, using LCG leads to worse performance than the single best solver, indicating poor generalization to unseen instances. In their work, Zhang et al. (2024) claimed that their GNN-based method outperforms traditional approaches. However, there are several differences: the instance and solver composition differ, they used a SAT-specific GNN layer, whereas we employ a general-purpose GIN, and our baselines use the latest SATzilla 2024 features, which have shown significantly better results than the previous version (Shavit & Hoos, 2024). Since the authors did not publish their code, we were unable to include their method as a baseline in our study.

Among the three baselines that are based on hand-crafted features,we observe that multi-class classification underperforms both pairwise classification and regression, reaching a negative closed gap on the small and medium-sized datasets. This highlights that algorithm selection is not a simple multi-class classification problem: in that formulation, the model loses valuable information about the performance of algorithms other than the best one. Finally, we find more gap closed for the large dataset across all three baselines, as it includes harder instances. When solving times increase, the benefit of algorithm selection grows, since feature computation accounts for a smaller fraction of the total computation time. Moreover, for harder instances, multiple solvers frequently time out, so choosing a solver that successfully solves the instance becomes especially valuable, given the additional penalty imposed by the PAR score.

Table 12: **SAT Solving.** Algorithm selection results (gap closed; higher is better).

| Instances | Method | VG | VCG | LCG |
|-----------|--------|-----|------|-----|
| SMALL | GIN | $0.05 \pm 0.00$ | $0.00 \pm 0.00$ | $-0.02 \pm 0.02$ |
| | PC | – | $0.03 \pm 0.02$ | – |
| | PR | – | $0.48 \pm 0.03$ | – |
| | MC | – | $-0.32 \pm 0.02$ | – |
| MEDIUM | GIN | – | – | – |
| | PC | – | $0.13 \pm 0.02$ | – |
| | PR | – | $0.44 \pm 0.02$ | – |
| | MC | – | $-0.13 \pm 0.02$ | – |
| LARGE | GIN | – | – | – |
| | PC | – | $0.33 \pm 0.01$ | – |
| | PR | – | $0.54 \pm 0.02$ | – |
| | MC | – | $0.23 \pm 0.02$ | – |

### B.3.2 COMBINATORIAL OPTIMIZATION: LEARNING TO PREDICT OPTIMAL SOLUTION OBJECTIVES

*Combinatorial optimization* (CO) lies at the intersection of optimization, operations research, discrete mathematics, and computer science (Korte & Vygen, 2012). It plays a central role in many practical applications, including vehicle routing, scheduling, and resource allocation (Paschos, 2014), underlining the importance of solving such optimization problems effectively in practice. CO problems involve selecting an optimal subset from a finite set to optimize an objective function subject to constraints.

Although CO problems are often computationally hard due to their discrete and non-convex nature (Karp, 1972), they can usually be solved effectively in practice using exact (Korte & Vygen, 2012), heuristic (Boussaïd et al., 2013), or approximation (Vazirani, 2010) methods. Classical research has focused on individual problem instances. However, in many real-world applications, structurally similar problems occur repeatedly. This motivates learning-based or data-driven approaches (Bengio et al., 2021; Cappart et al., 2021; Gasse et al., 2022), which aim to generalize across families of related instances.

Graphs play a central role in CO due to the inherently discrete structure of many problems and the prevalence of network data. Problems such as vehicle routing and the independent set problem are naturally expressed on graphs. Among the 21 NP-complete problems identified by Karp (1972), ten are graph problems; others, such as set covering, can also be formulated in graph terms. Additionally, variable-constraint relationships often induce bipartite graphs, where variables are connected to the constraints in which they appear. These structural patterns provide strong inductive biases for graph learning.

Formally, a *combinatorial optimization problem* is defined by a tuple $I = (\Omega, F, w)$, where

- $\Omega$ is a *finite set*,
- $F \subseteq 2^{\Omega}$ is the set of *feasible solutions*, and
- $w \colon \Omega \to \mathbb{R}$ assigns *weights* to elements of $\Omega$.

The *cost* or *objective value* of a solution $S \in F(I)$ is given by $c(S) = \sum_{\omega \in S} w(\omega)$. The goal is to find an optimal solution $S_I^* \in F(I)$ minimizing $c$ over the feasible set.[4]

**Related work**  Many recent learning-based optimization work rely on synthetic data (Toenshoff et al., 2021; Tönshoff et al., 2023; Zhang et al., 2023; Karalias & Loukas, 2020), either RB graphs (Xu et al., 2007), Erdős-Rényi (ER) graphs (Erdős & Rényi, 1960), or Barabási-Albert (BA) graphs (Albert & Barabási, 2002). Some also utilize pre-existing datasets from earlier graph benchmarks, e.g., (Karalias & Loukas, 2020) uses Morris et al. (2020), and Li et al. (2022) uses Planetoid datasets (Yang et al., 2016).

Earlier efforts have also produced benchmarks for specific graph CO problems. For graph coloring, examples include hard 3-colorability datasets featuring cliques (Mizuno & Nishihara, 2007; 2008), queen graphs (Chvátal, 2004), Latin square graphs (Gomes & Shmoys, 2002), $K$-Insertions graphs (Caramia & Dell'Olmo, 2002a), and $K$-FullIns graphs (Caramia & Dell'Olmo, 2002b). The DIMACS benchmark collection (Rossi & Ahmed, 2015) provides hundreds of real-world networks and problem instances across various CO domains. Still, their heterogeneity and limited number of instances per category make them less suited for modern machine learning pipelines. More recently, Nath & Kuhnle (2024) introduced a max-cut benchmark covering a variety of graph classes.

Our motivation differs from these prior works. We aim to design a unified set of graph datasets shared across multiple CO tasks, enabling direct comparison between methods in a consistent experimental setting. While some prior works generate random instances for individual problems, we standardize the process by fixing both the generation procedures and the train validation splits, ensuring that all methods compare on the same instances.

**Description of the learning tasks**  We include tasks for both supervised and unsupervised learning settings. For the supervised setting, the objective is to train a graph learning model to predict the optimal objective value of given CO instances. Given a set of CO problem instances $\mathcal{I}$ with corresponding optimal solutions $\mathcal{S}$, and a model $m \colon \mathcal{I} \to \mathbb{R}$, the learning task is to minimize the mean absolute error between the predicted and true objective values,

$$ 1/|\mathcal{I}| \sum_{I \in \mathcal{I}} |m(I) - c(S_I^*)| . \tag{3} $$

While this setup focuses on predicting the objective value rather than producing explicit solutions, it offers a practical advantage: it avoids the challenge of defining a meaningful loss function when multiple distinct optimal solutions exist. This simplification enables consistent, well-defined evaluation and learning.

In the unsupervised setting, we provide a differentiable surrogate loss function $\mathcal{L} \colon \mathbb{R}^{|\Omega(I)|} \to \mathbb{R}$ as well as decoders $d \colon \mathbb{R}^{|\Omega(I)|} \to F(I)$ for each CO problem. The model $m \colon \mathcal{I} \to \mathbb{R}^{|\Omega(I)|}$ is trained in an unsupervised fashion to predict a score for each variable that indicates whether it belongs to the solution set, minimizing

$$ 1/|\mathcal{I}| \sum_{I \in \mathcal{I}} \mathcal{L}(m(I)). \tag{4} $$

At test time, the model's output scores are converted into a feasible solution to the CO problem using the decoder. The model's performance is measured based on the objective value of the decoded solution,

---

[4]Without loss of generality, we assume minimization.

$c(d(m(I)))$. This setup is widely used in graph learning for CO literature (Karalias & Loukas, 2020; Min et al., 2022; Wenkel et al., 2024). We adapt our surrogate loss functions and decoders from this existing literature.

**Details on the dataset**  We consider three representative hard combinatorial optimization problems on graphs, the *maximum independent set* (MIS), the *max-cut*, and the *graph coloring problem*. These problems span a broad spectrum of structural properties and computational challenges.

Given a graph $G = (V(G), E(G))$, the MIS problem is to find the largest subset of non-adjacent nodes, formally,

- $\Omega := V(G)$
- $F := \{S \subseteq V(G) \mid \forall u, v \in S, (u, v) \notin E(G)\}$
- $w \colon V(G) \to \mathbb{R}$, in the unweighted case $w(v) = -1$

The objective is to find $S_I^* \in F$ minimizing $c(S) = \sum_{v \in S} w(v)$.

The max-cut problem seeks a 2-way partition of the node set such that the total weight of edges across the cut is maximized:

- $\Omega := E(G)$
- $F := \{S \subseteq E(G) \mid \exists V'(G) \subset V(G), \forall (u, v) \in S \colon u \in V(G), v \in V(G) \setminus V'(G)\}$
- $w \colon E(G) \to \mathbb{R}$, in the unweighted case $w(e) = -1$

The objective is to find $S_I^* \in F$ minimizing $c(S) = \sum_{e \in S} w(e)$.

Graph coloring is the problem of assigning at most $k$ colors to the nodes such that adjacent nodes receive different colors. Inspired by the assignment-based model (Jabrayilov & Mutzel, 2018), it can be formulated in search of a coloring function $f \colon V(G) \to [k]$, such that

- $\Omega := [k]$
- $F := \{S \subseteq [k] \mid \forall (u, v) \in E(G) \colon f(v) \neq f(u) \wedge \forall v \in V(G) \colon f(v) \in S\}$
- $w \colon \Omega \to \{1\}$

The objective is to find the best $S_I^* \in F$, such that the minimum number of colors is used, $c(S) = \sum_{s \in S} w(s)$.

To construct the datasets, we first define how the input graphs are generated. We consider three well-established random graph models: RB graphs (Xu et al., 2007), Erdős-Rényi (ER) graphs (Erdős & Rényi, 1960), and Barabási-Albert (BA) graphs (Albert & Barabási, 2002). These models are widely adopted in machine learning for solving combinatorial optimization problems.

The diversity and difficulty of each dataset are controlled via graph generation parameters. Specifically:

- For RB graphs, we specify integers $n$ (number of cliques), $k$ (nodes per clique), and a float $p$ (constraint tightness). The total number of nodes is $kn$, and we discard graphs whose size falls outside a predefined range.
- For BA graphs, we control the number of nodes and the parameter $m$, which determines how many edges are attached from each new node.
- For ER graphs, we use the number of nodes and the edge probability $p$, which controls the probability that an edge exists between any given pair of nodes.

We generate 50 000 instances for each dataset and provide both small-scale and large-scale variants. Table 13 shows a summary of the parameters used for graph generation. For RB graphs, parameters are taken from Zhang et al. (2023); Sanokowski et al. (2024); Wenkel et al. (2024); Sun et al. (2022). Parameters for large ER graphs are from Böther et al. (2022); Qiu et al. (2022); Sun & Yang (2023); Yu et al. (2024). Parameters for large BA graphs are from Böther et al. (2022). Parameters for ER-small and BA-small are adapted to match the graph size of RB-small. When a parameter is specified as a range, it is sampled uniformly at random for each instance. The only exception to this is the number of nodes in RB graphs, which is determined as described above.

Table 13: Parameters for CO problems generation.

| Generator | Size | Nodes | Nodes/Clique | Cliques | Tightness | Edge Prob. | Attached Edges |
|---|---|---|---|---|---|---|---|
| RB | small | [200, 300] | [5, 12] | [20, 25] | [0.3, 1] | - | - |
| | large | [800, 1200] | [20, 25] | [40, 55] | [0.3, 1] | - | - |
| ER | small | [200, 300] | - | - | - | 0.15 | - |
| | large | [700, 800] | - | - | - | 0.15 | - |
| BA | small | [200, 300] | - | - | - | - | 2 |
| | large | [700, 800] | - | - | - | - | 2 |

We provide datasets in both PYTORCH GEOMETRIC format (Fey & Lenssen, 2019; Fey et al., 2025) and NETWORKX format (Hagberg & Conway, 2020). Each instance retains only its adjacency matrix, and no node features are included.

For all the instances from various datasets, we use KaMIS (Lamm et al., 2016) for heuristic MIS solutions. The solution to Max-Cut is obtained by formulating it as an integer programming problem and solving it with Gurobi Optimization, LLC (2024), with a timeout of 3600 seconds. We use GCol (Lewis & Palmer, 2025) for a heuristic graph chromatic number, with a maximum of $10^8$ search iterations. The labeled datasets in our benchmark support both supervised learning, e.g., formulating a regression task to predict objective values, and unsupervised learning, where the goal is to find high-quality feasible solutions without access to ground-truth labels. The generality of the $(\Omega, F, w)$ framework makes it applicable to a wide range of combinatorial optimization problems, including all those proposed in our benchmark.

**Evaluation** For the supervised setting, we benchmark our baselines on the maximum independent set (MIS) problem, as defined in Appendix B.3.2. The task is to predict the CO objective value for the MIS problem in a supervised learning setting. We train all models using the mean absolute error (MAE) as the loss function and report the final test MAE for evaluation.

For the unsupervised setting, we evaluate on all three CO problems: MIS, maximum cut, and graph coloring. The models are trained in an unsupervised fashion using a problem-specific surrogate loss function. At test time, the model's output scores are converted into a feasible solution to the CO problem using a decoder. The CO problem's objective function is then used as a metric to measure the quality of the solutions obtained this way.

We conduct experiments on small and large synthetic graphs from the RB, ER, and BA families. We compare four baseline models: GIN, GT, MLP, and DeepSets. We use small-scale configurations for GIN, MLP, and DeepSets (with fixed depth and width), as well as a 5M parameter version of the GT. The exact hyperparameters used are listed in Table 26. Our proposed model utilizes the encoder-processor-decoder framework. The encoder, an MLP, processes node features derived from RWSE (Dwivedi et al., 2022b) and node degrees. The processor module is the only part that differs between baselines and performs the bulk of the calculations. It is followed by a decoder that generates node representations. For the supervised setting, we apply maximum aggregation to obtain a scalar graph-level prediction as the final output. For the unsupervised setting, we leave the node-level scores as the output.

**Results** The complete results for the supervised learning task are presented in Table 14. Among the baselines, GIN achieves the best performance across most datasets. We attribute this to the strong inductive bias of MPNN, which aligns well with the structure of graph-based CO problems. The MLP baseline generally underperforms DeepSet, suggesting that global information aggregation is beneficial for this task, except for the ER graphs. Conversely, the GT performs poorly on some datasets. We hypothesize that this is due to the training difficulties associated with this task.

Table 15 shows the baseline results for unsupervised CO. We report the average objective value of solutions obtained by running the decoder on the model's output scores, for problem instances in the test set. Out of the four baselines, GIN performs best across all three CO problems, with some exceptions. As expected, the graph-based models GIN and GT generally perform better than MLP and DeepSet. One exception is graph coloring on the (very sparse) BA graphs, where DeepSet outperforms the other two models on both graph sizes. Note that our baselines do not utilize diffusion or other advanced architectures or techniques tailored explicitly for CO. Their performance is therefore much weaker compared to methods that do use

them, e.g. Sanokowski et al. (2024); Zhang et al. (2023). Also note that most learning-based methods currently cannot compete with exact solvers and some hand-crafted heuristics in terms of solution quality.

Table 14: **Combinatorial Optimization.** Results for each CO dataset with supervised learning (MIS).

| Dataset | Method | Size | |
|---|---|---|---|
| | | **Small** | **Large** |
| RB graph | GIN | $0.491 \pm 0.099$ | $2.125 \pm 0.484$ |
| | GT | $4.112 \pm 2.353$ | $0.915 \pm 0.235$ |
| | MLP | $1.583 \pm 0.052$ | $1.437 \pm 0.520$ |
| | DeepSet | $0.918 \pm 0.186$ | $1.427 \pm 0.224$ |
| ER graph | GIN | $0.234 \pm 0.191$ | $0.352 \pm 0.265$ |
| | GT | $6.486 \pm 8.101$ | $9.641 \pm 15.335$ |
| | MLP | $0.751 \pm 0.767$ | $0.914 \pm 0.307$ |
| | DeepSet | $0.756 \pm 0.711$ | $2.244 \pm 0.364$ |
| BA graph | GIN | $0.292 \pm 0.041$ | $0.111 \pm 0.016$ |
| | GT | $3.481 \pm 3.446$ | $1.829 \pm 1.383$ |
| | MLP | $2.825 \pm 0.949$ | $3.383 \pm 0.504$ |
| | DeepSet | $2.304 \pm 0.262$ | $3.362 \pm 1.491$ |

### B.3.3 ALGORITHMIC REASONING: LEARNING GRAPH INVARIANTS

**Motivation for inclusion in GraphBench**

**Related work**   Algorithmic reasoning has been adopted for various architectures  (Li et al., 2024a; Velickovic et al., 2020a). Apart from neural networks  (Diao & Loynd, 2023; Mahdavi et al., 2023; Bounsi et al., 2024; Rodionov & Prokhorenkova, 2023) and reinforcement learning approaches  (Estermann et al., 2024),  Velickovic et al. (2022) propose the CLRS benchmark for a variety of algorithmic tasks, derived from the *Introduction to Algorithms* textbook  (Cormen et al., 2009). This benchmark comprises 30 different algorithmic problems, accompanied by hints to guide the solution of each task. However, this benchmark does not include regression tasks and restricts learning tasks to specific algorithms. Furthermore, follow-up works investigated the importance of hints given in CLRS  (Rodionov & Prokhorenkova, 2023) and proposed a variety of architectures for solving algorithmic tasks (Bevilacqua et al., 2023; Ibarz et al., 2022; Li et al., 2024a).

**Description of learning task**   We consider a collection of seven algorithmic reasoning tasks, selected from classical graph problems studied in algorithmics. These tasks include computing the topological sorting  (Kahn, 1962), identifying bridges in a graph, computing the minimum spanning tree (Kruskal, 1956), determining the maximum flow (Goldberg & Tarjan, 1986), finding the maximum clique, computing Steiner trees (Kou et al., 1981), and solving bipartite matching (Hopcroft & Karp, 1973).

Our task set spans different output granularities: node-level tasks (topological sorting, bipartite matching, max clique), edge-level tasks (bridges, minimum spanning tree, Steiner tree), and a graph-level task (max flow). To introduce additional modeling challenges, we provide three levels of difficulty per task, representing variations in the underlying graph distributions.

Each task is formulated as either a binary classification or a regression problem. Given a set of graphs $\mathcal{G}_A$, an algorithmic reasoning task $A$, its ground-truth solution function $S_A$, and a model $m$, the goal is to train

$$m \colon \mathcal{G}_A \to \mathbb{R} \quad \text{or} \quad m \colon \mathcal{G}_A \to [0, 1],$$

depending on the task. The objective is to minimize the mean absolute error (MAE) between the model's predictions and the ground-truth solution:

$$\frac{1}{|\mathcal{G}_A|} \sum_{G \in \mathcal{G}_A} |m(G) - S_A(G)|.$$

Table 15: **Combinatorial Optimization.** Results for each CO dataset with unsupervised learning.

| Problem | Dataset | Method | Size Small | Size Large |
|---|---|---|---|---|
| MIS (MIS size ↑) | RB | GIN | 17.294± 0.328 | 13.999± 0.321 |
| | | GT | 16.542± 0.477 | 13.406± 0.140 |
| | | MLP | 16.105± 0.097 | 13.040± 0.214 |
| | | DeepSet | 16.021± 0.032 | 13.183± 0.035 |
| | | Solver | 20.803± 1.817 | 42.547± 4.449 |
| | ER | GIN | 25.418± 0.407 | 26.276± 0.408 |
| | | GT | 22.984± 0.473 | 24.980± 0.292 |
| | | MLP | 23.183± 0.016 | 24.259± 0.449 |
| | | DeepSet | 23.050± 0.061 | 24.220± 0.056 |
| | | Solver | 33.604± 1.428 | 45.637± 0.631 |
| | BA | GIN | 100.16± 3.674 | 135.00± 0.720 |
| | | GT | 99.579± 6.448 | 114.26± 0.601 |
| | | MLP | 95.108± 2.042 | 114.49± 0.758 |
| | | DeepSet | 95.076± 0.173 | 114.89± 0.016 |
| | | Solver | 142.86± 16.54 | 433.77± 19.17 |
| Maximum Cut (maximum cut size ↑) | RB | GIN | 2106.7± 14.62 | 24748.± 87.76 |
| | | GT | 1925.7± 32.75 | 21524.± 184.0 |
| | | MLP | 1727.7± 165.1 | 20357.± 249.6 |
| | | DeepSet | 140.02± 155.5 | 3575.9± 730.0 |
| | | Solver | 2920.1± 97.23 | 33914.± 7861. |
| | ER | GIN | 2327.9± 24.78 | 20878.± 107.9 |
| | | GT | 2172.7± 91.75 | 16534.± 278.0 |
| | | MLP | 1866.7± 67.64 | 7335.4± 57.49 |
| | | DeepSet | 33.634± 20.84 | 27.663± 6.763 |
| | | Solver | 2835.5± 607.6 | 23884.± 1809. |
| | BA | GIN | 397.00± 0.605 | 1044.1± 0.649 |
| | | GT | 363.76± 0.639 | 986.93± 3.128 |
| | | MLP | 308.73± 0.224 | 929.20± 4.060 |
| | | DeepSet | 1.0669± 0.800 | 154.31± 151.5 |
| | | Solver | 460.91± 50.13 | 1260.4± 48.81 |
| Graph Coloring (number of colors used ↓) | RB | GIN | 25.166± 0.288 | 55.513± 0.526 |
| | | GT | 25.146± 0.253 | 55.562± 0.648 |
| | | MLP | 24.733± 0.667 | 55.558± 0.557 |
| | | DeepSet | 26.723± 0.189 | 71.051± 0.604 |
| | | Solver | 19.970± 3.465 | 41.480± 6.634 |
| | ER | GIN | 16.182± 0.202 | 34.587± 0.545 |
| | | GT | 16.188± 0.201 | 34.385± 0.413 |
| | | MLP | 17.110± 0.144 | 34.658± 0.394 |
| | | DeepSet | 18.266± 0.018 | 55.345± 0.551 |
| | | Solver | 10.235± 0.836 | 22.933± 0.772 |
| | BA | GIN | 5.1318± 0.114 | 6.2028± 0.283 |
| | | GT | 5.0939± 0.070 | 6.1167± 0.086 |
| | | MLP | 5.9900± 0.127 | 9.4215± 0.186 |
| | | DeepSet | 3.2780± 0.122 | 3.1981± 0.239 |
| | | Solver | 3.0000± 0.000 | 3.0000± 0.000 |

For binary classification tasks, we instead report the F1 score computed from the objective to maximize the accuracy between predictions and ground truth:

$$\frac{1}{|\mathcal{G}_A|} \sum_{G \in \mathcal{G}_A} \mathbb{1}(m(G) = S_A(G))$$

where $\mathbb{1}$ denotes the indicator function.

**Details on the dataset** For all algorithmic reasoning tasks, we work with sets of synthetically generated graphs. We begin by sampling graphs using a variety of graph generators, including Erdős–Rényi (ER) graphs (Erdős & Rényi, 1960), stochastic block model (SBM) graphs (Holland et al., 1983), power-law cluster (PC) graphs (Holme & Kim, 2002), Newman–Watts–Strogatz (NWS) graphs (Newman & Watts, 1999), Barabási–Albert (BA) graphs (Barabási & Albert, 1999), and dual Barabási–Albert (DBA) graphs (Moshiri, 2018). During the sampling process, we ensure that each resulting graph is unique; otherwise, it is resampled.

For each task, we choose generator parameters shown in Table 16 to yield meaningful distributions of task-specific properties. We further introduce the parameter of component connections, determining the number of random edges between disconnected components in a graph. This allows us to sample graphs with low edge probabilities without disconnected components. Additionally, based on the difficulty level (as described in the previous section), we adjust the probability distribution over the selected graph generators to further control the complexity of generated instances, as seen in Table 17. For minimum spanning tree computations, we additionally shift the graph generator parameters as indicated in Table 16 to require additional generalization capabilities, as the same parameters proved to be too easy to generalize to. Ground-truth labels are computed using the NETWORKX library (Hagberg & Conway, 2020), which provides reference implementations of most of the required graph algorithms.

Table 16: Graph generation parameters for algorithmic reasoning tasks. The cc parameter denotes the number of connections between disconnected components in Erdős–Rényi graphs.

| Generator | Param. | Top. Sort | MST | MST (shift) | Bridges | Steiner Trees | Max. Clique | Flow | Max. Matching |
|---|---|---|---|---|---|---|---|---|---|
| ER | p | 0.3 | 0.19 | 0.17 | 0.11 | 0.14 | 0.9 | 0.16 | 0.08 |
| | cc | 1 | 1 | 1 | 1 | 1 | 1 | 1 | 1 |
| NWS | k | 2 | 4 | 2 | - | 4 | 4 | 4 | 6 |
| | p | 0.2 | 0.2 | 0.15 | - | 0.12 | 0.6 | 0.2 | 0.18 |
| BA | m | 2 | 3 | 2 | 1 | 2 | 8 | 3 | 3 |
| DBA | $m_1$ | 3 | 4 | 2 | 3 | 3 | 4 | 4 | 4 |
| | $m_2$ | 2 | 2 | 1 | 1 | 1 | 2 | 2 | 2 |
| | p | 0.5 | 0.3 | 0.05 | 0.07 | 0.4 | 0.3 | 0.3 | 0.2 |
| PC | p | 0.1 | 0.4 | 0.35 | 0.5 | 0.7 | 0.5 | 0.5 | 0.5 |
| | m | 3 | 1 | 5 | 1 | 3 | 9 | 5 | 8 |
| SBM | p. mat. | $\begin{bmatrix} 0.5 & 0.2 \\ 0.2 & 0.5 \end{bmatrix}$ | $\begin{bmatrix} 0.4 & 0.4 \\ 0.4 & 0.4 \end{bmatrix}$ | $\begin{bmatrix} 0.31 & 0.01 \\ 0.01 & 0.31 \end{bmatrix}$ | $\begin{bmatrix} 0.5 & 0.01 \\ 0.01 & 0.5 \end{bmatrix}$ | $\begin{bmatrix} 0.4 & 0.4 \\ 0.4 & 0.4 \end{bmatrix}$ | $\begin{bmatrix} 0.75 & 0.75 \\ 0.75 & 0.75 \end{bmatrix}$ | $\begin{bmatrix} 0.35 & 0.3 \\ 0.3 & 0.35 \end{bmatrix}$ | $\begin{bmatrix} 0.31 & 0.1 \\ 0.1 & 0.31 \end{bmatrix}$ |
| | sizes | 1/2, 1/2 | 1/2, 1/2 | 1/2, 1/2 | 1/2, 1/2 | 1/2, 1/2 | 1/2, 1/2 | 1/2, 1/2 | 1/2, 1/2 |

Table 17: Graph generation parameters for algorithmic reasoning tasks.

| Split | Difficulty | ER | PC | NWS | BA | DBA | SBM |
|---|---|---|---|---|---|---|---|
| Train | EASY | 1/3 | 0 | 1/3 | 0 | 1/3 | 0 |
| | MEDIUM | 1 | 0 | 0 | 0 | 0 | 0 |
| | HARD | 1 | 0 | 0 | 0 | 0 | 0 |
| Valid/Test | EASY | 1/6 | 1/6 | 1/6 | 1/6 | 1/6 | 1/6 |
| | MEDIUM | 1/6 | 1/6 | 1/6 | 1/6 | 1/6 | 1/6 |
| | HARD | 0 | 1/5 | 1/5 | 1/5 | 1/5 | 1/5 |

For edge-level tasks, we also generate an edge-level representation to support models that require tokenized input.

Each task includes one million generated training graphs and 10 000 validation and test graphs. To assess generalization beyond function approximation, we fix the number of nodes in training graphs to 16, while setting validation and test graphs to have 128 nodes.

To further study generalization, we introduce size generalization tasks, where pre-trained models are evaluated on graphs with sizes ranging from 128 to 512 nodes. We exclude max flow computation from size generalization as the computation of the MAE scales with the number of nodes present in the flow network and is therefore not indicative of size generalization. This setup enables an empirical analysis of how well models extrapolate to unseen graph sizes. Except for the topological sorting, Steiner tree, and bipartite matching tasks, no node features are given. For the minimum spanning tree, max flow, and Steiner tree tasks, corresponding edge weights are included as edge attributes.

**Evaluation** We use the encoder-processor-decoder pipeline described in Appendix D. For node and graph-level tasks, the pipeline is applied directly with the task-specific encoder described below. Since the graph transformer baseline expects node-level tokenization, we provide an additional graph transform. We leverage the following edge transform which allows us to provide node level tokenization on a modified graph $G'$ obtained from the original graph $G$, with $V(G') := \{(v, v) \mid v \in V(G)\} \cup E(G)$ and $E(G') := \{((u, v), (w, z)) \mid u = w \lor u = z \lor v = w \lor v = z\}$.

Since most of our tasks only provide either node or edge features, we use learned vectors as described in the encoder for the remaining embeddings. Across all datasets, we employ one-layer linear embeddings for both integer and real-valued features.

Depending on the node, edge, or graph level tasks, the decoder applies linear layers that represent the classification or regression target. In the case of bridges, MST, Steiner trees, max clique and bipartite matching a binary classification target is used for node or edge level predictions. For flow and topological sorting, a one-dimensional regression target is used instead.

For pretrained models used in size generalization experiments, we use the model directly as encoder, processor, and decoder. However, the input graph size is increased. In contrast to complete training examples, we conduct a few-shot scenario with only 1000 graphs to allow for fast inference. In addition, we do not consider the flow task here due to its regression setting. Throughout all experiments, we use the same random seed for a pretrained model to enable comparison to the 128-node test set.

**Results** We report results on all datasets for each of the three selected difficulties. These are denoted using the terms EASY, MEDIUM, and HARD to represent the additional generalization requirements from out-of-distribution sampling. For classification tasks, we report F1 scores, and for regression tasks, we report MAE. All results are grouped by their respective estimated task difficulty in Table 18 and Table 19 below. Additionally, we report size generalization results using pre-trained baselines from the MEDIUM setting of our training procedure. For this, we evaluate each pre-trained model, starting at a graph node count of 128 and up to 512. Results are provided in Table 20.

For MST, max. Clique and max matching datasets show improved performance when using the graph transformer baseline. However, for max flow, bridges, and Steiner tree datasets, the GIN baseline model performs better throughout the proposed difficulty levels. Furthermore, we observe a reduction in F1 scores with increasing difficulty across datasets, highlighting the varying generator selections in the training data. However, the differences are not as pronounced as expected for the bridges dataset.

We further observe that the GIN baseline is inconsistent across seeds for MST and Steiner tree tasks, whereas the graph transformer baseline does not suffer from similar issues. Overall, the performance of both baselines is the lowest for Steiner trees and max clique tasks, indicating that these tasks are more complex for graph learning baselines to learn. Similar to the results obtained in the CLRS benchmark (Velickovic et al., 2022) and by Müller et al. (2024b) for an expressive graph transformer architecture, MST, bridges, and topological sorting provide more manageable tasks for graph learning baselines to solve. We note that despite previous benchmarks such as CLRS, there currently exist no other baseline results on the synthetic algorithmic reasoning datasets we provide.

In the case of size generalization, we observe that both baselines are relatively robust to the size scaling of the test graphs. However, we notice that the size generalization capability is task-dependent, with MST and Steiner tree tasks improving with larger graph sizes. Nonetheless, we note that this behavior may be observed due to the shift in average degree and node connectivity, using the same parameters as in test set generation, as every graph generator method selected scales with the number of nodes. Nonetheless, performance is decreasing for topological sorting, max matching and max clique tasks, highlighting the incapability of our selected baselines to be size generalization invariant across all tasks.

Table 18: Results for algorithmic reasoning datasets including topological sorting, minimum spanning tree, bridge finding, and Steiner tree computation.

| Task | Difficulty | Model | MAE | F1 |
|---|---|---|---|---|
| Topological Sorting | EASY | GIN | $0.1001 \pm 0.0089$ | – |
| | | GT | $0.116 \pm 0.0058$ | – |
| | MEDIUM | GIN | $0.1537 \pm 0.0031$ | – |
| | | GT | $0.1305 \pm 0.0094$ | – |
| | HARD | GIN | $0.1301 \pm 0.0046$ | – |
| | | GT | $0.1532 \pm 0.0214$ | – |
| MST | EASY | GIN | – | $0.6906 \pm 0.1655$ |
| | | GT | – | $0.8566 \pm 0.0068$ |
| | MEDIUM | GIN | – | $0.7288 \pm 0.0894$ |
| | | GT | – | $0.8504 \pm 0.0148$ |
| | HARD | GIN | – | $0.6107 \pm 0.2015$ |
| | | GT | – | $0.8421 \pm 0.0115$ |
| Bridges | EASY | GIN | – | $0.9831 \pm 0.0184$ |
| | | GT | – | $0.9269 \pm 0.0103$ |
| | MEDIUM | GIN | – | $0.9622 \pm 0.0077$ |
| | | GT | – | $0.8762 \pm 0.0258$ |
| | HARD | GIN | – | $0.968 \pm 0.0178$ |
| | | GT | – | $0.8897 \pm 0.0304$ |
| Steiner Trees | EASY | GIN | – | $0.6691 \pm 0.0288$ |
| | | GT | – | $0.6691 \pm 0.0112$ |
| | MEDIUM | GIN | – | $0.5628 \pm 0.1100$ |
| | | GT | – | $0.5672 \pm 0.0790$ |
| | HARD | GIN | – | $0.5516 \pm 0.2368$ |
| | | GT | – | $0.5212 \pm 0.0219$ |

Table 19: Results for algorithmic reasoning datasets, including max clique, flow, and max matching.

| Task | Difficulty | Model | MAE | F1 |
|------|------------|-------|-----|-----|
| Max Clique | EASY | GIN | – | $0.4584 \pm 0.0101$ |
| | | GT | – | $0.5407 \pm 0.0088$ |
| | MEDIUM | GIN | – | $0.3996 \pm 0.0852$ |
| | | GT | – | $0.4859 \pm 0.0017$ |
| | HARD | GIN | – | $0.4102 \pm 0.0820$ |
| | | GT | – | $0.4868 \pm 0.0013$ |
| Flow | EASY | GIN | $3.4387 \pm 0.0631$ | – |
| | | GT | $4.2737 \pm 0.0646$ | – |
| | MEDIUM | GIN | $9.5960 \pm 0.1707$ | – |
| | | GT | $6.3786 \pm 0.4262$ | – |
| | HARD | GIN | $9.5061 \pm 0.1265$ | – |
| | | GT | $6.4833 \pm 0.0869$ | – |
| Max Matching | EASY | GIN | – | $0.7527 \pm 0.0051$ |
| | | GT | – | $0.7402 \pm 0.0172$ |
| | MEDIUM | GIN | – | $0.6399 \pm 0.0231$ |
| | | GT | – | $0.6915 \pm 0.009$ |
| | HARD | GIN | – | $0.6595 \pm 0.0164$ |
| | | GT | – | $0.6743 \pm 0.0038$ |

Table 20: Size generalization results on algorithmic reasoning datasets from Section 3.3.3. Each column represents an evaluation on 1000 graphs with the given number of nodes. For each experiment, the same generation parameters were used as in the results presented in Table 18 and Table 19. OOT denotes the case where computation of the underlying size generalization data was not completed in less than 24 hours of compute on a single cluster node, as detailed in Appendix D. All generated graphs are based on the MEDIUM difficulty setting. We use the same single seed for each pretrained model from each task to provide inference results on different sizes.

| Dataset (Score) | Model | 128 | 192 | 256 | 384 | 512 |
|-----------------|-------|-----|-----|-----|-----|-----|
| Topological Sorting (MAE) | GT | 0.1346 | 0.1730 | 0.1827 | 0.2018 | 0.2071 |
| | GIN | 0.1492 | 0.181 | 0.1981 | 0.2015 | 0.2179 |
| MST (F1) | GT | 0.8720 | 0.8773 | 0.8891 | 0.8874 | 0.8817 |
| | GIN | 0.6103 | 0.8132 | 0.8605 | 0.8765 | 0.8823 |
| Bridges (F1) | GT | 0.8799 | 0.8842 | 0.8959 | 0.9049 | 0.9118 |
| | GIN | 0.9579 | 0.9213 | 0.9190 | 0.9203 | 0.9212 |
| Steiner Trees (F1) | GT | 0.5160 | 0.5322 | 0.5221 | 0.5762 | 0.5578 |
| | GIN | 0.5499 | 0.5739 | 0.6338 | 0.6651 | 0.6502 |
| Max Clique (F1) | GT | 0.4877 | 0.4849 | 0.4890 | 0.4915 | 0.4931 |
| | GIN | 0.3496 | 0.3112 | 0.3148 | 0.2926 | 0.2673 |
| Max Matching (F1) | GT | 0.7010 | 0.6552 | 0.6206 | 0.5770 | OOT |
| | GIN | 0.6271 | 0.6326 | 0.6372 | 0.6382 | OOT |

## B.4 EARTH SYSTEMS

### B.4.1 WEATHER FORECASTING: MEDIUM-RANGE ATMOSPHERIC STATE PREDICTION

**Related work** The idea of utilizing machine learning models for weather forecasting first emerged in 2020, when Rasp et al. (2020) and Weyn et al. (2020) employed CNN architectures to predict global weather at resolutions of 5.625° and 1.9°, respectively. A key breakthrough occurred when Keisler (2022) first applied MPNNs to represent the globe more naturally, enabling the prediction of 3D states 6 hours ahead and facilitating multi-day forecasts. This approach has already achieved a skill comparable to that of 1° global NWP (GFS/ECMWF) on specific metrics, outperforming prior data-driven models.

Building on this idea, Lam et al. (2023) introduced *GraphCast*, a multi-scale MPNN-based system that delivers today's forecasts at 0.25° resolution. GraphCast achieved unprecedented accuracy, outperforming even ECMWF's *high-resolution deterministic model* (HRES) out to ten days.

The success of these deterministic forecasts raised the question of probabilistic prediction. Price et al. (2025) introduced *GenCast*, a generative weather model that produces an ensemble of forecasts rather than a single deterministic run. For this, they used a diffusion-based approach built on an MPNN backbone, outperforming ECMWF's *ensemble system* (ENS), the world's top operational probabilistic forecast on over 97% of evaluated metrics.

Several other advances can be noted. Oskarsson et al. (2024) have proposed Graph-EFM, a model featuring a hierarchical GNN that handles global and regional forecasts seamlessly. With *OneForecast*, Gao et al. (2025) have introduced an MPNN framework using nested multi-scale graphs and adaptive message passing to improve local extreme event predictions within a global context. These works indicate a trend toward hybrid solutions that densify graphs in target regions for high-resolution detail while maintaining global consistency.

Current state-of-the-art models, such as Pangu-Weather (Bi et al., 2022), Aurora (Bodnar, 2024), and FGN (Alet et al., 2025), all present excellent results that, in many areas, far exceed those of conventional NWP methods.

It is important to note, however, that while machine learning-based approaches exceed NWP skill on tested time scales, they do not explicitly enforce conservation laws. Over long integration horizons, this can lead to subtle physical drifts that may limit specific climate-related uses. This becomes particularly relevant under regime shifts, for example, caused by climate change, that lie outside the distributions the models were trained on. Furthermore, compared to NWP models, ML-based approaches offer very limited interpretability, as they function mainly like black boxes.

**Description of the learning task** The objective of the task is to model medium-range weather evolution by predicting the residual change in the atmospheric state over a fixed time horizon of twelve hours. Specifically, given an initial snapshot of the current atmospheric state, the model forecasts the twelve-hour future change in meteorological variables.

To achieve this, the initial normalized grid values, which contain atmospheric data from various pressure levels at each grid point, are mapped to learned node attributes on an icosahedral node mesh using a single MPNN layer, as described in GraphCast (Lam et al., 2023). Several rounds of message passing are then performed on this icosahedron, and the resulting states are mapped back to grid data using another single MPNN layer. This process yields a residual change in the atmospheric variables, which is then added to the original data to obtain an updated weather state twelve hours after the initial timestep.

The training objective is a spatially and variable-weighted *mean-squared error* for the twelve-hour-ahead prediction, inspired by the training object used in GraphCast (Lam et al., 2023). For each verification time $d \in D$, the model ingests $x_{d-2}$ and predicts a residual change $\Delta x$, yielding $\hat{x}^d = x_{d-2} + \Delta x$. The loss compares $\hat{x}^d$ with $x^d$. The loss function thus corresponds to

$$\mathcal{L}(x^d, \hat{x}^d) = \frac{1}{|D|\,|G|\,\sum_{j \in J} |L_j|} \sum_{d \in D} \sum_{i \in G} \sum_{j \in J} \sum_{\ell \in L_j} a_i\, w_j\, s_{j,\ell} \left( \hat{x}^d_{i,j,\ell} - x^d_{i,j,\ell} \right)^2,$$

with level weights

$$s_{j,\ell} = \begin{cases} \dfrac{P_\ell}{\frac{1}{|L_j|} \sum_{m \in L_j} P_m}, & \text{if } j \text{ is multi-level, e.g. atmospheric variables,} \\ 1, & \text{if } j \text{ is single-level, e.g. surface variables.} \end{cases}$$

where $D$ is the set of forecast date-times, $G$ the grid cells, $J$ the variables, $L_j$ the pressure levels for variable $j$, $a_i = \dfrac{\cos(\text{lat}_i)}{\frac{1}{|G|}\sum_{k\in G}\cos(\text{lat}_k)}$ are mean-normalized latitude weights, $w_j$ are variable weights, and $P_\ell$ are the pressure levels. Predicting a residual change $\Delta x$ rather than an absolute value improves stability and generalization (He et al., 2020).

For the evaluation, we report an unweighted mean-squared error for each variable $j$.

$$\text{MSE}_j(x^d, \hat{x}^d) = \frac{1}{|D|\,|G|\,|L_j|}\sum_{d\in D}\sum_{i\in G}\sum_{\ell\in L_j}\left(\hat{x}^d_{i,j,\ell} - x^d_{i,j,\ell}\right)^2,$$

and an unweighted mean-squared error over all variables.

$$\text{MSE}_{\text{all}}(x^d, \hat{x}^d) = \frac{1}{|D|\,|G|\,\sum_{j\in J}|L_j|}\sum_{d\in D}\sum_{i\in G}\sum_{j\in J}\sum_{\ell\in L_j}\left(\hat{x}^d_{i,j,\ell} - x^d_{i,j,\ell}\right)^2.$$

It should be noted here that the success of a weather forecast can be measured in many ways other than just the mean-squared error. For example, *WeatherBench 2* (Rasp et al., 2024) provides a comprehensive overview of various metrics and their relevance.

In addition to a 12-hour forecast, it is possible to predict the weather autoregressively over more extended periods of time. For example, the 12-hour forecast can be used as input for another 12-hour forecast. GraphCast (Lam et al., 2023) employs a six-step process, i.e., a three-day forecast. GenCast (Price et al., 2023) can even generate weather forecasts for up to 15 days.

**Details on the dataset**   We utilize reanalysis data from the ERA5 dataset, which has been preprocessed via the WeatherBench2 pipeline (Rasp et al., 2024). Multiple resolutions of this dataset are available. We use a downsampled version containing a $64 \times 32$ equiangular grid, employing a conservative area-preserving interpolation. ERA5 data has a temporal resolution of six hours, with timesteps at 0h, 6h, 12h, and 18h. Each weather state contains 62 physical and derived variables, 15 of which are defined across the 13 pressure levels and 47 are defined on the surface. Of those, we only use five variables for the surface and six for the atmospheric levels each, as done by Price et al. (2025). The pressure-level variables include quantities like temperature, humidity, and wind speeds. The surface-level variables also include more general information about the location, such as a land-sea mask, sea level pressure, and precipitation.

We provide a dataset with the aforementioned resolution of $64 \times 32$ from ERA5, processed by the WeatherBench2 pipeline. The dataset is available as a PYTORCH GEOMETRIC data object. It includes the original grid data in the form of a 3D graph, the icosahedron as a mesh graph, and the mapping edges between the two. The weights of this mapping must be learned during training, but the mappings themselves are fixed and can therefore be provided as static information.

The 3D grid graph contains 2 048 nodes, each linked by a directed edge to one of the 2 562 mesh nodes of the icosahedron. Together, these two structures form a single graph of 4 610 nodes connected by 59 667 edges. Each grid node contains information about five surface variables and six atmospheric variables across 13 pressure levels ranging from 50hPa to 1000hPa. For the exact variables, see Table 21.

**Evaluation**   We use an encoder-processor-decoder pipeline as described in Appendix D. While the encoder and the decoder leverage edge features for their respective mappings, we will only evaluate the final prediction, which comes in the form of node features in the final Earth's coordinate graph, where we compute the MSE between the predicted atmospheric state of $t + 2$ and the true observed state $t + 2$ from the data, over all variables and timesteps.

**Results**   We provide results for each weather variable at three different pressure levels, aligning with the results obtained in WeatherBench2 (Rasp et al., 2024), which allows for an easily accessible comparison of weather forecasting models with traditional weather forecasting baselines. For comparative models like GraphCast and Persistence, only selected results are available. Thus, we compare against those in particular. However, we also provide cumulative errors across all pressure levels present in the underlying data. Throughout our evaluation, we provide a 12-hour forecast prediction across all variables.

Our baselines establish a transparent and reproducible lower bound for medium-range weather forecasting. While they do not reach the performance of Persistence or specialized systems like GraphCast, this is

expected given their simpler architecture, shorter training time, and deliberately straightforward setup. Several factors contribute to this outcome. Unlike specialized systems, our models were trained for substantially fewer iterations, with fewer computing resources and without extensive hyperparameter optimization. We deliberately prioritized transparency and reproducibility over complexity in the architectural design. Our training setup is intentionally simple and does not incorporate domain-specific refinements, such as positional edge features, which are known to improve forecast skill. Moreover, medium-range weather forecasting is a particularly demanding task that often benefits from specialized evaluation protocols, whereas our assessment relies on a basic, uniform metric.

Importantly, the purpose of this baseline is not to achieve state-of-the-art performance. Its main contribution lies in providing a transparent and reproducible lower bound that defines a precise reference point against which future graph learning methods can be compared.

Table 21: Mean squared error for each weather variable for selected pressure levels, the cumulative error for each atmospheric variable over all pressure levels, and the cumulative error over all variables and pressure levels. The selected pressure levels align with the evaluated levels of WeatherBench2 (Rasp et al., 2024). Persistence denotes a basic weather forecasting model, which provides a forecast by simply assuming that variable values remain constant, as they are in the input values. GraphCast refers to the GraphCast model proposed by Lam et al. (2023). Table formatting k indicates a multiplicative factor of $10^3$ while n indicates $10^{(-9)}$.

| Pressure Level | Variable | GT | Persistence | GraphCast |
|---|---|---|---|---|
| All | All variables | 179.629k | - | - |
| Surface | 2-m temperature (2T) | 7.572 | 7.123 | 0.068 |
| | 10-m u wind component (10U) | 5.072 | 2.166 | 0.012 |
| | 10-m v wind component (10V) | 6.102 | 3.266 | 0.013 |
| | Mean sea level pressure (MSL) | 102.551k | 60.056k | 240.832 |
| | Total precipitation (TP) | 0.009 | 714.517n | 52.377n |
| 500 | Temperature (T) | 2.751 | 1.120 | 0.007 |
| | U component of wind (U) | 15.026 | 6.658 | 0.048 |
| | V component of wind (V) | 27.003 | 12.988 | 0.053 |
| | Geopotential (Z) | 86.250k | 48.637k | 155.057 |
| | Specific humidity (Q) | 0.010 | 66.211n | 1.090n |
| | Vertical wind speed (W) | 0.032 | 6.242 | 0.050 |
| 700 | Temperature (T) | 2.406 | 1.051 | 0.006 |
| | U component of wind (U) | 8.716 | 3.951 | 0.025 |
| | V component of wind (V) | 14.156 | 6.754 | 0.027 |
| | Geopotential (Z) | 51.669k | 32.773k | 141.296 |
| | Specific humidity (Q) | 0.009 | 250.556n | 3.304n |
| | Vertical wind speed (W) | 0.024 | 3.352 | 0.027 |
| 850 | Temperature (T) | 2.832 | 1.351 | 0.009 |
| | U component of wind (U) | 9.151 | 4.015 | 0.022 |
| | V component of wind (V) | 12.289 | 6.565 | 0.023 |
| | Geopotential (Z) | 51.542k | 32.453k | 139.716 |
| | Specific humidity (Q) | 0.010 | 354.058n | 4.881n |
| | Vertical wind speed (W) | 0.016 | 3.517 | 0.024 |
| All | Temperature (T) | 2.805 | - | - |
| | U component of wind (U) | 13.855 | - | - |
| | V component of wind (V) | 21.501 | - | - |
| | Geopotential (Z) | 77.020k | - | - |
| | Specific humidity (Q) | 0.010 | - | - |
| | Vertical wind speed (W) | 0.018 | - | - |

## C HARDWARE

### C.1 SAT SOLVING

We ran the SAT solvers on a cluster equipped with two AMD EPYC 7 543 32-core processors, each with 512MB of L3 cache and 1 TB of RAM, running Rocky Linux 9. We allowed each solver to use up to 64GB of memory, and therefore up to 8 solvers were run on the same node in parallel. Before conducting the experiments, we performed a preliminary experiment, running Kissat on 100 random SAT instances 10 times, ensuring consistent results (in terms of computation time).

For training the graph neural networks, we used a cluster with 4 GPU nodes, each containing 4 H100 SXM GPUs with 80GB of vRAM.

### C.2 ALGORITHMIC REASONING

Each algorithmic reasoning task was evaluated on a cluster node equipped with one NVIDIA L40 GPU, featuring 48GB of vRAM and 512GB of RAM. Throughout all experiments, the same cluster configuration was used to allow for comparability of wall clock times. Due to computation time constraints, the computation of positional and structural encodings (such as RWSE and LPE) or edge transformations in the case of the graph transformer baseline was provided beforehand.

### C.3 WEATHER FORECASTING

For weather forecasting, a cluster node with four NVIDIA L40 GPUs was used during evaluation. Additionally, a cluster node with four NVIDIA A100 GPUs was utilized during training to accommodate larger batch sizes. Both cluster nodes use 512GB of RAM. All wall clock times are obtained using the L40 cluster node. In contrast to algorithmic reasoning experiments, the positional encodings are computed at computation time.

### C.4 COMBINATORIAL OPTIMIZATION

Experiments for unsupervised CO were performed on an NVIDIA H100 GPU with 80GB of vRAM, and for supervised learning were on an NVIDIA L40S GPU with 48 GB vRAM.

## D EXPERIMENTAL SETUP

Similar to Bechler-Speicher et al. (2025), we provide an encoder-processor-decoder architecture for all tasks, which we detail in the following. While task-specific parts of our architecture vary, we offer a general architecture to measure the performance of selected baselines on our tasks. In the case of the chip design and weather forecast dataset, we adapt this architecture to provide the necessary task-dependent encoder and decoder functionality.

**Encoder** For each task, we provide a task-specific encoder that derives encodings for node- and edge-level features with a common dimension $d \in mathbbR^+$. These features are then passed to the processor using node and or edge features. For models that require tokenized input, we provide node-level tokens. For missing node or edge features, a learnable vector of dimension $d$ is used in their place. In case of the graph transformer baseline, we add a `[cls]` token for graph-level representations, following Bechler-Speicher et al. (2025).

**Processor** As our processor, we select one of the baseline models we evaluate. These consist of two dummy baselines: an MLP and a DeepSet architecture, a GNN baseline given by a GIN architecture, and a Graph Transformer baseline. Independent of the selected baseline, we compute updated node and graph representations at each layer as follows, given a graph $G$ with node representations $\boldsymbol{X}$,

$$\boldsymbol{X} = \phi(\mathsf{LayerNorm}(\boldsymbol{X}), G) \qquad (5)$$

$$\boldsymbol{X} = \boldsymbol{X} + \mathsf{MLP}(\mathsf{LayerNorm}(\boldsymbol{X}))$$

with the following MLP layer:

$$\mathsf{MLP}(\boldsymbol{x}) := \sigma(\mathsf{LayerNorm}(\boldsymbol{x})\boldsymbol{W}_1)\boldsymbol{W}_2,$$

where $\sigma$ denotes the GeLU nonlinearity and $\boldsymbol{W}_1, \boldsymbol{W}_2$ are weight matrices. Further, $\phi$ is the selected baseline, and MLP denotes a two-layer multilayer perceptron with GELU non-linearity (Hendrycks & Gimpel, 2016). For the graph transformer baseline, we employed biased attention as described by Bechler-Speicher et al. (2025). Furthermore, we incorporate RWSE (Dwivedi et al., 2022b) and LPE (Müller & Morris, 2024) as node positional encodings for our GNN and graph transformer baselines.

**Decoder** Since every task requires task-specific predictions, we provide a decoder for each. Throughout all functions, we use the same decoder layout, differing only in the predictions made. We propose a decoder two-layer multilayer perceptron for our baseline models consisting of two linear layers $\boldsymbol{W}_1 \in \mathbb{R}^{d \times d}, \boldsymbol{W}_2 \in \mathbb{R}^{d \times o}$ as well as GELU non-linearity, with $o$ denoting the task-specific output dimension, i.e.,

$$\boldsymbol{W}_2(\mathsf{LayerNorm}(\mathsf{GELU}(\boldsymbol{W}_1 x))).$$

Optionally, a bias term can be added to the decoder.

**Pretraining** In addition to training models using our encoder-processor-decoder architecture, we provide support for task evaluation of pretrained models for specific tasks. These are further specified in each subsection concerning additional evaluation benchmarks.

**Evaluation protocol** Using the aforementioned hyperparameter tuning process, we evaluate all baseline models across three random seeds for each task and report their mean performance along with the standard deviations. For out-of-distribution generalization tasks, we utilize pretrained models from these baseline evaluations.

Moreover, we report metrics such as memory usage, wall-clock time for training, and inference latency to provide insights into the real-world applications of our benchmark tasks.

## D.1 HYPERPARAMETER SELECTION

Here, we provide the hyperparameters used in training our baseline models for each task. We note that in most cases, no extensive hyperparameter search was conducted for the baseline models.

Table 22: List of hyperparameters used for baseline models in BlueSky datasets.

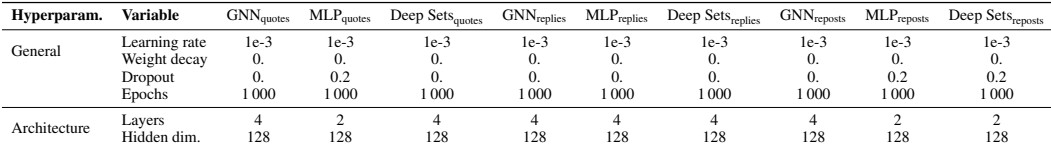

| Hyperparam. | Variable | GNN$_{\text{quotes}}$ | MLP$_{\text{quotes}}$ | Deep Sets$_{\text{quotes}}$ | GNN$_{\text{replies}}$ | MLP$_{\text{replies}}$ | Deep Sets$_{\text{replies}}$ | GNN$_{\text{reposts}}$ | MLP$_{\text{reposts}}$ | Deep Sets$_{\text{reposts}}$ |
|---|---|---|---|---|---|---|---|---|---|---|
| General | Learning rate | 1e-3 | 1e-3 | 1e-3 | 1e-3 | 1e-3 | 1e-3 | 1e-3 | 1e-3 | 1e-3 |
| | Weight decay | 0. | 0. | 0. | 0. | 0. | 0. | 0. | 0. | 0. |
| | Dropout | 0. | 0.2 | 0. | 0. | 0. | 0. | 0. | 0.2 | 0.2 |
| | Epochs | 1 000 | 1 000 | 1 000 | 1 000 | 1 000 | 1 000 | 1 000 | 1 000 | 1 000 |
| Architecture | Layers | 4 | 2 | 4 | 4 | 4 | 4 | 4 | 2 | 2 |
| | Hidden dim. | 128 | 128 | 128 | 128 | 128 | 128 | 128 | 128 | 128 |

Table 23: List of hyperparameters used for baseline models in SAT solving tasks. These include parameters for the choice of positional encoding (PE), architecture (GIN), and general learning parameters.

| Hyperparam. | Variable | GIN$_{\text{EPM: VG}}$ | GT$_{\text{EPM: VCG}}$ | GIN$_{\text{EPM: LCG}}$ | GIN$_{\text{AS: VG}}$ | GT$_{\text{AS: VCG}}$ | GIN$_{\text{AS: LCG}}$ |
|---|---|---|---|---|---|---|---|
| General | Learning rate | 1e-3 | 1e-3 | 1e-5 | 1e-5 | 1e-5 | 1e-5 |
| | Batch size | 16 | 16 | 16 | 16 | 16 | 16 |
| | Weight decay | 0.1 | 0.1 | 0.1 | 0.1 | 0.1 | 0.1 |
| | Dropout | 0.0 | 0.0 | 0.0 | 0.0 | 0.05 | 0.0 |
| | Steps | 100 000 | 100 000 | 100 000 | 100 000 | 100 000 | 100 000 |
| Optimizing | Warmup steps | 10 000 | 10 000 | 10 000 | 10 000 | 10 000 | 10 000 |
| | Scheduler | | | CosineAnnealingLR | | | |
| | Optimizer | | | AdamW, betas=(0.9,0.999) | | | |
| | Param. | 2.2M | 2.2M | 2.2M | 2.2M | 2.2M | 2.2M |
| PE | Type | None | None | None | None | None | None |
| | Encoding dim. | 384 | 384 | 384 | 384 | 384 | 384 |
| | Hidden dim. | 384 | 384 | 384 | 384 | 384 | 384 |
| Architecture | Layers | 6 | 16 | 6 | 16 | 6 | 16 |
| | Hidden dim. | 384 | 384 | 384 | 384 | 384 | 384 |
| | Activation | GELU | GELU | GELU | GELU | GELU | GELU |

Table 24: List of hyperparameters used for baseline models in algorithmic reasoning tasks. These include parameters for the choice of positional encoding (PE), architecture (GT/GIN), and general learning parameters. The respective abbreviations denote the following tasks: topological sorting (TO), minimum spanning tree (MST), bridges (BR), and Steiner tree (ST).

| Hyperparam. | Variable | $GIN_{TO}$ | $GT_{TO}$ | $GIN_{MST}$ | $GT_{MST}$ | $GIN_{BR}$ | $GT_{BR}$ | $GIN_{ST}$ | $GT_{ST}$ |
|---|---|---|---|---|---|---|---|---|---|
| General | Learning rate | 1e-4 | 1e-4 | 3e-4 | 3e-4 | 2e-4 | 2e-4 | 1e-4 | 1e-4 |
| | Batch size | 256 | 256 | 256 | 256 | 256 | 256 | 256 | 256 |
| | Weight decay | 0.1 | 0.1 | 0.1 | 0.1 | 0.1 | 0.1 | 0.1 | 0.1 |
| | Dropout | 0.1 | 0.1 | 0.1 | 0.1 | 0.1 | 0.1 | 0.1 | 0.1 |
| | Steps | 11 880 | 11 880 | 11 880 | 11 880 | 11 880 | 11 880 | 11 880 | 11 880 |
| Optimizing | Warmup steps | 1 000 | 1 000 | 1 000 | 1 000 | 1 000 | 1 000 | 1 000 | 1 000 |
| | Scheduler | | | | CosineAnnealingLR | | | | |
| | Optimizer | | | | AdamW, betas=(0.9,0.999) | | | | |
| | Param. | 2.55M | 14.93M | 2.68M | 15.07M | 2.55M | 14.93M | 2.55M | 14.93M |
| PE | Type | RWSE | RWSE | LPE | LPE | RWSE | RWSE | RWSE | RWSE |
| | Num. steps/Eigvals | 16 | 16 | 32 | 32 | 16 | 16 | 16 | 16 |
| | Encoding dim. | 384 | 384 | 384 | 384 | 384 | 384 | 384 | 384 |
| | Hidden dim. | 768 | 768 | 384 | 384 | 768 | 768 | 768 | 768 |
| | Num. enc. layers | 2 | 2 | 2/2 | 2/2 | 2 | 2 | 2 | 2 |
| Architecture | Layers | 6 | 16 | 6 | 16 | 6 | 16 | 6 | 16 |
| | Hidden dim. | 384 | 384 | 384 | 384 | 384 | 384 | 384 | 384 |
| | Attn. heads | 0 | 16 | 0 | 16 | 0 | 16 | 0 | 16 |
| | Activation | GELU | GELU | GELU | GELU | GELU | GELU | GELU | GELU |

Table 25: List of hyperparameters used for baseline models in algorithmic reasoning and weather forecasting tasks. These include parameters for the choice of positional encoding (PE), architecture (GT/GIN), and general learning parameters. The following tasks are denoted by the respective abbreviations: max clique (MC), flow (FL), maximum matching (MM), and weather forecasting (WE).

| Hyperparam. | Variable | $GIN_{MC}$ | $GT_{MC}$ | $GIN_{FL}$ | $GT_{FL}$ | $GIN_{MM}$ | $GT_{MM}$ | $GT_{WE}$ |
|---|---|---|---|---|---|---|---|---|
| General | Learning rate | 1e-4 | 1e-4 | 1e-4 | 1e-4 | 1e-4 | 1e-4 | 1e-4 |
| | Batch size | 256 | 256 | 256 | 256 | 256 | 256 | 16 |
| | Weight decay | 0.1 | 0.1 | 0.1 | 0.1 | 0.1 | 0.1 | 0.1 |
| | Dropout | 0.1 | 0.1 | 0.1 | 0.1 | 0.1 | 0.1 | 0.1 |
| | Steps | 11 880 | 11 880 | 11 880 | 11 880 | 11 880 | 11 880 | 8 000 |
| Optimizing | Warmup steps | 1 000 | 1 000 | 1 000 | 1 000 | 1 000 | 1 000 | 500 |
| | Scheduler | | | | CosineAnnealingLR | | | |
| | Optimizer | | | | AdamW betas=(0.9,0.999) | | | |
| | Param. | 2.68M | 15.07M | 2.55M | 14.93M | 2.55M | 14.93M | 15.35M |
| PE | Type | LPE | LPE | RWSE | RWSE | RWSE | RWSE | None |
| | Num. steps/Eigvals | 16 | 16 | 32 | 32 | 16 | 16 | 0 |
| | Encoding dim. | 384 | 384 | 384 | 384 | 384 | 384 | 384 |
| | Hidden dim. | 384 | 384 | 768 | 768 | 768 | 768 | 768 |
| | Num. enc. layers | 2/2 | 2/2 | 2 | 2 | 2 | 2 | 2 |
| Architecture | Layers | 6 | 16 | 6 | 16 | 6 | 16 | 16 |
| | Hidden dim. | 384 | 384 | 384 | 384 | 384 | 384 | 384 |
| | Attn. heads | 0 | 16 | 0 | 16 | 0 | 16 | 16 |
| | Activation | GELU | GELU | GELU | GELU | GELU | GELU | GELU |

Table 26: List of hyperparameters used for baseline models in CO tasks. These include parameters for the choice of positional encoding (PE), architecture (GT/GIN), and general learning parameters.

| Hyperparam. | Variable | GIN | GT | MLP | DeepSet |
|---|---|---|---|---|---|
| General | Learning rate | 1e-3 | 1e-3 | 1e-3 | 1e-3 |
| | Batch size | 256 | 256 | 256 | 256 |
| | Weight decay | 0. | 0. | 0. | 0. |
| | Dropout | 0. | 0. | 0. | 0. |
| | Epochs | 1 000 | 1 000 | 1 000 | 1 000 |
| | Patience | 100 | 100 | 100 | 100 |
| Optimizing | Warmup steps | 0 | 0 | 0 | 0 |
| | Optimizer | Adam | Adam | Adam | Adam |
| | Param. | 2.1M | 5.6M | 1.2M | 2.1M |
| PE | Type | RWSE | RWSE | RWSE | RWSE |
| | Num. steps/Eigvals | 16 | 16 | 16 | 16 |
| | Encoding dim. | 384 | 384 | 384 | 384 |
| | Hidden dim. | 384 | 384 | 384 | 384 |
| | Num. enc. layers | 2 | 2 | 2 | 2 |
| Architecture | Layers | 6 | 6 | 6 | 6 |
| | Hidden dim. | 384 | 384 | 384 | 384 |
| | Attn. heads | 0 | 4 | 0 | 0 |
| | Activation | GELU | GELU | GELU | GELU |

Table 27: List of hyperparameters used for baseline models in Electronic circuits tasks. These include parameters for the choice of architecture (GT/GIN/GCN/GAT) and general learning parameters.

| Hyperparam. | Variable | GT | GIN | GCN | GAT |
|---|---|---|---|---|---|
| General | Learning rate | 1e-3 | 1e-3 | 1e-3 | 1e-3 |
| | Batch size | 512 | 512 | 512 | 512 |
| | Weight decay | 0. | 0. | 0. | 0. |
| | Dropout | 0. | 0. | 0. | 0. |
| | Epochs | 7 00 | 7 00 | 7 00 | 7 00 |
| Architecture | Layers | 6 | 4 | 4 | 4 |
| | Hidden dim. | 384 | 384 | 384 | 384 |
| | Attn. heads | 0 | 4 | 0 | 0 |
| | Activation | GELU | RELU | RELU | RELU |

## D.2    BASELINE ARCHITECTURES

In the following, we provide implementation details on the baselines used for the datasets in GRAPHBENCH. For dataset-specific design choices, we provide detailed information in Appendix D.3.

**Graph transformer architecture**    As described in Appendix D we use an encoder-processor-decoder baseline across tasks. In this case, we consider a graph transformer as the processor architecture following the implementation outlined in Bechler-Speicher et al. (2025). For most tasks, we consider a node-level tokenization, where each node of a graph is treated as a single token input to the graph transformer. However, for edge-level tasks, we use the transformation outlined for algorithmic reasoning tasks, allowing for edge-level tokens to be used without changes to the processor architecture design. Additionally, absolute PEs, such as RWSE or LPE, are added to the node embeddings before the first graph transformer layer. Then, the graph transformer layer computes full multi-head scaled-dot-product attention, adding an attention bias $B$ to the unnormalized attention matrix and applying softmax to it. Let $Q, K, V \in \mathbb{R}^{L \times d}$ and $B \in \mathbb{R}^{L \times L}$ with $L$ denoting the number of tokens and $d$ the embedding dimension. Then attention and a graph transformer layer take the form

$$\text{Attention}(Q, K, V, B) := \text{softmax}\big(d^{-\frac{1}{2}} \cdot QK^T + B\big)V,$$

$$X^{t+1} := \text{MLP}\big(\text{Attention}(X^t W_Q, X^t W_K, X^t W_V, B)\big),$$

where $W_Q, W_K, W_V \in \mathbb{R}^{d \times d}$ are learnable weight matrices and MLP denotes a two layer-MLP. In practice, we compute attention over multiple heads, allowing for different attention biases to be added to the attention matrix. With $\phi$ given by the multihead attention computation, a processor layer is provided by Equation (5). In the processor, multiple layers are stacked together, allowing for the pipeline showcased in Appendix D.

**GIN architecture**    Throughout this work, we use a GINE-based graph neural network baseline as the processor in our outlined framework in Appendix D. Following, Hu et al. (2020b) the GINE layer updates the node representations $h_v^{(t)}$ at iteration $t$ as follows:

$$h_v^{(t+1)} = \text{N}\Big((1 + \epsilon)h_v^{(t)} + \sum_{u \in \mathcal{N}(v)} \text{ReLU}(h_v^{(t)} + e_{uv})\Big)$$

where $\mathcal{N}(v)$ denotes the neighborhood of a node $v$ and N a neural network such as an MLP.

We apply the GINE processor layer in a similar way to the graph transformer baseline design outlined previously by replacing $\phi$ with a GINE layer where N is given by a dropout layer. First, the node embeddings, optionally with added PEs, are passed to the GINE layer, where layer normalization is applied. Then the output of the GINE message passing layer is forwarded to a two-layer MLP. We use the same residual connection as seen in the GIN implementation from Bechler-Speicher et al. (2025). We then stack multiple layers together to provide the processor part of our baseline architecture. Unless otherwise specified, we use mean pooling for graph-level tasks at the end of the processor step.

## D.3    ARCHITECTURE CHOICES FOR DATASETS

**Social Networks**    On the proposed BlueSky datasets, we opted for a different GNN architecture. The high(er) number of nodes, along with high average node degree and variance, pushed us towards resorting to a message-passing architecture aggregating messages over neighborhoods via averaging (instead of summation as in GIN). We opted, in particular, for a variant of GraphConv with mean aggregation. Node representations $h_v^{(t)}$ are, namely, updated as follows:

$$h_v^{(t+1)} = \sigma\Big(W_1^{(t)} h_v^{(t)} + \frac{1}{\deg_{in}(v)} \sum_{u:\ u \to v} W_2^{(t)} h_u^{(t+1)}\Big), \tag{6}$$

where $\sigma$ is set to ReLU, dropout is applied before its application and, finally, no normalization layers are interleaved.

# E AUTOMATED HYPERPARAMETER OPTIMIZATION

GRAPHBENCH also integrates automated hyperparameter optimization (HPO), which aims to easily tune new models, improve performance, and enhance reproducibility. The hyperparameter optimization in GRAPHBENCH is based on the SMAC3 (Lindauer et al., 2022) package, which utilizes multi-fidelity scheduling. This approach evaluates many configurations on lower budgets and eliminates those with poor performance. As graph learning approaches are expensive to train, this approach enables the saving of costly resources. Additionally, SMAC3 utilizes a surrogate model to propose new configurations, thereby learning from previous experience and further reducing the computational load.

Due to limited computational resources, we were unable to run HPO on all datasets and domains. To showcase the effectiveness of the automated hyperparameter optimization included in GRAPHBENCH, we tuned the GIN model on the SMALL SAT dataset with VG. We set the total optimization budget to 150 evaluations, varying fidelity over training gradient steps between 1 000 and 100 000.

The used configuration space is shown in Table 28, and the results are presented in Table 29. We observe a 73% improvement of the RMSE value between the manually tuned version and the automatically tuned version, demonstrating that the automated HPO in GRAPHBENCH can be beneficial for improving model performance.

Table 28: GIN Configuration space used in the hyperparameter optimisation experiment.

| Hyperparameter | Range |
|---|---|
| Learning rate | [1e-6, 0.1] |
| Weight decay | [1e-8, 0.1] |
| Warmup iters | [1 000, 20 000] |
| Dropout | [0.0, 0.5] |

Table 29: **SAT Solving.** RMSE ($\log_{10}$ time) for performance prediction on SMALL SAT instances. We compare the manually tuned version of GIN to the automatically tuned version. Lower is better.

| Solver | Method | VG |
|---|---|---|
| KISSAT | GIN | $1.36\pm_{0.15}$ |
| | GIN – Tuned | $1.26\pm_{0.03}$ |

