# OpenReview forum: "GraphBench: Next-generation graph learning benchmarking"
_ICLR.cc/2026/Conference — ICLR 2026 Conference Desk Rejected Submission_

### Official Review · Reviewer_TXkq · 2025-10-21

**Soundness:** 2
**Presentation:** 2
**Contribution:** 2
**Rating:** 2
**Confidence:** 4

**Summary:**

The paper introduces GraphBench, a unified benchmarking suite for graph learning that spans multiple domains (social networks, hardware and chip design, reasoning and optimization, and earth systems) and supports node-, edge-, graph-level, and generative tasks under standardized splits and domain-relevant metrics, with explicit tests for out-of-distribution generalization. It releases datasets and a Python package with PyTorch Geometric–compatible loaders, fixed train/validation/test splits, and a common hyperparameter-tuning/evaluation pipeline to promote reproducibility. The authors provide baselines using message-passing neural networks and graph transformers and report cross-domain observations. Overall, the contributions are the curated multi-domain dataset suite, standardized protocols and metrics with OOD evaluation, reference baselines, and accessible software intended to catalyze more robust and comparable graph learning research.

**Strengths:**

The paper’s strengths are mostly infrastructural: it assembles a single benchmark that spans multiple domains and task regimes, and bakes in out-of-distribution evaluations rather than treating them as an afterthought. On originality, the contribution is less about a new task than about scope and unification—moving beyond molecule/citation staples to include social networks, chip and circuit design, SAT algorithm selection/performance prediction, and weather, with node-, edge-, graph-level, and generative tasks under one roof. On quality, the suite specifies fixed splits, domain-relevant metrics, hyperparameter tuning scripts, and a common evaluation pipeline, and it runs baselines across multiple seeds—choices that encourage reproducibility. In terms of significance, the inclusion of large, practically motivated datasets (for example, a SAT benchmark with more than 100k instances) and explicit temporal/size splits makes this a potentially useful reference point for the community.

**Weaknesses:**

The main weaknesses are on evaluation depth and model coverage: despite the breadth of tasks, the benchmark tests only a small set of generic baselines (essentially GIN/GINE and one graph transformer, plus MLP/DeepSets), trained with minimal tuning and only three seeds, which limits what we can conclude about method rankings and robustness; broadening to stronger modern variants and running more seeds with proper HPO across domains would materially improve credibility. In SAT, core baselines are constrained by engineering choices: the graph transformer is excluded due to positional-encoding cost and out-of-memory issues, clause graphs are dropped, and evaluation is restricted to small formulas, so the domain’s hardest settings are not probed; meanwhile, classic tabular feature models outperform the provided GNNs by a wide margin, underscoring that the current baselines may not reflect best available practice. For algorithmic reasoning, the paper itself notes seed instability for GIN on several tasks, reinforcing the need for more repetitions and variance-aware reporting. For weather, the baseline substantially underperforms simple persistence and purpose-built systems like GraphCast and is intentionally simplified, which is understandable for a first release but reduces the benchmark’s immediate diagnostic value; adding stronger domain baselines and task-relevant metrics beyond a single MSE would help.

**Questions:**

- Model coverage feels too narrow for the benchmark’s ambitions; can you expand beyond GIN/GINE, a single graph transformer, MLP, and DeepSets to include stronger, task-specific and modern graph baselines (e.g., CO methods the paper itself notes outperform simple baselines, and additional GT variants)?

- Baseline robustness and tuning seem under-probed: results are averaged over only three seeds and automated HPO is demonstrated on a single case; could you increase seeds (e.g., 10+ where feasible), report confidence intervals/significance tests, and roll out a budgeted, uniform HPO protocol so method rankings are more reliable across domains?

---

> ### Author Response · Authors · 2025-11-21
>
> We thank the reviewer for their detailed feedback and constructive suggestions. Please find our responses below:
>
> Weaknesses:
>
> >The main weaknesses are on evaluation depth and model coverage: despite the breadth of tasks, the benchmark tests only a small set of generic baselines (essentially GIN/GINE and one graph transformer, plus MLP/DeepSets), trained with minimal tuning and only three seeds, which limits what we can conclude about method rankings and robustness; broadening to stronger modern variants and running more seeds with proper HPO across domains would materially improve credibility.
>
> We appreciate the reviewer’s concerns regarding the depth of evaluation and breadth of model coverage. Our intention with this benchmark paper was to provide strong initial baselines using well-established models, including GIN/GINE, a representative graph transformer based on theoretical expressiveness, MLPs, and DeepSets, across a diverse set of tasks. This approach follows standard practice in benchmark releases, where the community is encouraged to build upon these results with more specialized or state-of-the-art methods (as seen in OGB, LRGB, and other benchmarks). Notably, our selected graph transformer baseline has demonstrated robust performance across domains ([1]).
>
> While we agree that expanding model variants, increasing seed counts, and implementing comprehensive hyperparameter optimization would strengthen credibility, such extensive experimentation was not feasible given our computational resources (of an academic lab). Our focus has been on establishing an extensible foundation for future research rather than exhaustive benchmarking at this stage.
>
> >In SAT, core baselines are constrained by engineering choices: the graph transformer is excluded due to positional-encoding cost and out-of-memory issues, clause graphs are dropped, and evaluation is restricted to small formulas, so the domain’s hardest settings are not probed
>
> We respectfully disagree that the exclusion of Graph Transformers (GTs) on large graphs limits the validity of the evaluation. In the context of SAT solving, efficiency is not just a feature; it is the objective. The computational cost of positional encodings for GTs and the memory requirements for large clause graphs for GIN make their adoption in many use-cases (such as solving medium and large size formulas) impractical.
>
> Our evaluation demonstrates that while simpler, hand-crafted feature baselines scale effectively, heavy-weight GNN or GT architectures hit a 'scalability wall.' Thus, the inability to run GTs on large instances is, in itself, an important result: it confirms that current quadratic-complexity architectures are ill-suited for the 'hardest settings' of the SAT domain. We provide large-scale benchmarks to encourage future work on graph models that can actually compete in this rigorous environment.
>
> >For algorithmic reasoning, the paper itself notes seed instability for GIN on several tasks, reinforcing the need for more repetitions and variance-aware reporting
>
> We acknowledge the reviewer’s observation of seed instability in GIN across several algorithmic reasoning tasks. In our view, this instability indicates the inherent difficulty these tasks pose for standard GNN architectures such as GIN. To provide a more comprehensive understanding of performance variability, we intend to conduct additional runs with a larger number of seeds and report both the variance and the mean for these tasks until the end of the discussion period. We then would incorporate these results into the paper.
>
> Furthermore, we recognize that a theoretical examination of such instabilities would be a valuable direction for future research.
>
> [1] Generalizable Insights for Graph Transformers in Theory and Practice, Stoll et al., NeurIPS 2025

---

> ### Author Response · Authors · 2025-11-21
>
> >For weather, the baseline substantially underperforms simple persistence and purpose-built systems like GraphCast and is intentionally simplified, which is understandable for a first release but reduces the benchmark’s immediate diagnostic value; adding stronger domain baselines and task-relevant metrics beyond a single MSE would help.
>
> With respect to the weather forecasting benchmark, our approach was to establish baseline performance using a generic graph transformer architecture rather than domain-specific models. We have included comparisons with both GraphCast and simple persistence baselines to contextualize our results. It is important to note that purpose-built systems like GraphCast are typically trained on higher-resolution data over more extended periods and with substantially greater computational resources, as documented in [2] and [3]. This difference in experimental setup naturally leads to improved performance by those specialized models.
>
> To further enhance transparency, we are open to incorporating results from other published machine learning models trained on comparable datasets where available. Additionally, while our primary evaluation metric is MSE, we can supplement this with MAE scores for completeness. We note that this MAE would be unweighted, unlike the weighted MSE usually reported.  It should also be noted that many advanced metrics used in recent domain-specific evaluations (such as those assessing extreme weather events) require supplementary data not currently included in our release. Running experiments on large-scale weather datasets also presents significant computational challenges compared to other domains within our benchmark.
>
> These clarifications contextualize our methodological choices while highlighting avenues for future improvements, such as the integration of stronger domain-specific baselines and more diverse evaluation metrics as the benchmark evolves.
>
> Questions:
>
>
> >Model coverage feels too narrow for the benchmark’s ambitions; can you expand beyond GIN/GINE, a single graph transformer, MLP, and DeepSets to include stronger, task-specific and modern graph baselines (e.g., CO methods the paper itself notes outperform simple baselines, and additional GT variants)?
>
> We agree that expanding model coverage could further strengthen benchmarking efforts; however, due to dataset scale and resource limitations, we prioritized widely-used baselines (GIN/GINE), which have demonstrated strong performance on various benchmarks in previous works [4][5]. The selected graph transformer variant was chosen based on its theoretical evaluation of performance and applicability without relying on specific architecture choices [1]. In addition we select Deep Set and MLP baselines to highlight the value and importance of connectivity in the proposed datasets.
>
> For future releases or follow-up work, we plan to broaden baseline coverage further where feasible.
> Nonetheless, even with current baselines, our results highlight significant trends: e.g., cases where classical methods outperform graph-based approaches on certain tasks.
>
> >Baseline robustness and tuning seem under-probed: results are averaged over only three seedand automated HPO is demonstrated on a single case; could you increase seeds (e.g., 10+ where feasibles ), report confidence intervals/significance tests, and roll out a budgeted, uniform HPO protocol so method rankings are more reliable across domains?
>
> Given the number and size of datasets included in this benchmark, running every experiment with a high number of seeds (e.g., 10+) or rolling out uniform automated HPO protocols across all tasks was unfortunately not feasible within available resources (of an academic lab). Automated hyperparameter optimization was therefore demonstrated on a single SAT dataset as an illustrative case study only.
>
> We will increase seed counts for select experiments in the camera-ready version. It is worth noting that most benchmark/dataset papers do not routinely report confidence intervals or significance tests across all tasks due to similar practical constraints; nonetheless, we agree that these are valuable directions for future extensions.
>
> Please let us know if you have any remaining concerns. Please consider updating your score if you are satisfied with our responses. We are happy to answer additional questions.
>
> [1] Generalizable Insights for Graph Transformers in Theory and Practice, Stoll et al., NeurIPS 2025
>
> [2] GraphCast: Learning skillful medium-range global weather forecasting, Lim et al., Science 2022
>
> [3]GenCast: Diffusion-based ensemble forecasting for medium-range weather, Price et al., Nature 2025
>
> [4] Can Classic GNNs Be Strong Baselines for Graph-level Tasks? Simple Architectures Meet Excellence, Luo et al., ICML 2025
>
> [5] Where Did the Gap Go? Reassessing the Long-Range Graph Benchmark, Tönshoff et al., LoG 2023

---

### Official Review · Reviewer_k5Hy · 2025-10-30

**Soundness:** 2
**Presentation:** 1
**Contribution:** 1
**Rating:** 2
**Confidence:** 3

**Summary:**

This paper introduces GraphBench, a contribution of around 20 unique datasets from 7 broad and diverse categories for graph learning benchmarking. It complements the existing popular graph learning benchmarks which may be significant for molecular and citation networks, as examples, but often missing for other areas such as chip design, circuit design and weather forecasting, among others (though there are individual areas in the literature that tackle these problems). The paper also highlights the current limitations with graph benchmarks in terms of data diversity reflecting multiple real world scenarios, in/out distribution splits, evaluation consistencies and framework for usage. It finally presents a framework based on Pytorch and Pytorch Geometric which acts as the interface for loaders, optimizers and evaluators.

**Strengths:**

- The scope, size and diversity of datasets in the proposed benchmark is comprehensive, as compared to existing benchmarks such as OGB.
- Some of the domains, eg. algorithmic reasoning, has benchmarks but require different interfacing. This paper addresses practical gaps like these.
- Each dataset consists of elaborate details on motivations, statistics, intra-category diversity with multiple unique datasets, alongside baseline MPNN and GT models.
- The benchmark can be used with a user-friendly python package standard with other graph and other modalities' benchmarks, and can be integrated in existing graph learning workflows conveniently.

**Weaknesses:**

- The manuscript includes reasonable discussion points on limitations of existing benchmarks, however its contribution is limited as a dataset paper, without addressing major prior limitation points. For instance, it does not explore correlations or transfer patterns between domains (e.g., how models that perform well in reasoning tasks transfer in physical or social domains), although size generalization experiments are included.
- Other evaluations are primarily empirical baseline (MPNN, GT) scores which follow the literature.
- Many of the limitations with graph benchmarks as mentioned above in summary sections are well known and corresponding remedies are implemented - for eg. OGB has consistent evaluators, realistic splits, and other benchmarks also follow these or part of these characteristics.
- Writing inconsistencies- some datasets have their experiments in main paper, while some have been pushed to appendix.
- There are references which may be incorrect and could be a result of LLM-assisted writing or search.

**Questions:**

na

---

> ### Author Response · Authors · 2025-11-21
>
> We thank the reviewer for their thorough evaluation and thoughtful comments. Please find our detailed responses to each point below:
>
> >The manuscript includes reasonable discussion points on limitations of existing benchmarks, however its contribution is limited as a dataset paper, without addressing major prior limitation points. For instance, it does not explore correlations or transfer patterns between domains (e.g., how models that perform well in reasoning tasks transfer in physical or social domains), although size generalization experiments are included.
>
> We appreciate the reviewer’s recognition of our discussion on the limitations of existing benchmarks.
> The aggregation and release of large-scale datasets spanning seven underrepresented domains in graph machine learning is itself a significant contribution. Other major benchmarks (e.g., OGB) also initially focused on data curation and baseline results, with more comprehensive analyses following in subsequent work.
>
> Our benchmark lays the groundwork for such future studies by providing both data and an extensible framework; we envision this as an ongoing project open to further expansion by us and the community. GraphBench is one of the most diverse and extensive benchmarking efforts in the graph learning community.
>
> We acknowledge that deeper analysis, such as exploring correlations or transfer patterns between domains, would be of interest to the community. However, due to computational constraints (of an academic lab), it is not feasible for us to perform extensive cross-domain transfer experiments at this time.
> Additionally, we will also expand our discussion of known limitations for graph learning methods in specific domains with additional references from recent literature.
>
>
> >Other evaluations are primarily empirical baseline (MPNN, GT) scores which follow the literature.
>
> Our empirical evaluations primarily focus on established baselines (MPNN (GIN), GT) evaluated extensively in prior works. The number and diversity of baselines provided are in line with those used in other leading benchmarks such as OGB. We view the baseline as solid starting points. Additionally, our selected baselines demonstrated robust performance across domains in previous evaluations [1][2].
>
> In certain domains, such as weather forecasting or chip design, we recognize that domain-specific models may outperform general-purpose GNNs; however, implementing every advanced method across all tasks was beyond our current scope, given resource constraints. Our goal is to provide a strong foundation for others to build more specialized evaluations.
>
> >Many of the limitations with graph benchmarks, as mentioned above in summary sections, are well known and corresponding remedies are implemented - for eg, OGB has consistent evaluators, realistic splits, and other benchmarks also follow these or part of these characteristics.
>
> While many existing benchmarks have implemented remedies such as consistent evaluators or realistic splits (e.g., OGB), there remains a lack of variety in the domains covered, which we specifically address by introducing new datasets across seven domains, with multiple datasets in each.
>
> Additionally, benchmark updates are often inconsistent (e.g., missing fixes for torch.load in OGB) or limited to select domains; our framework is designed with systematic updates and broad applicability across multiple settings in mind.
>
>
> [1] Can Classic GNNs Be Strong Baselines for Graph-level Tasks? Simple Architectures Meet Excellence, Luo et al., ICML 2025
>
> [2] Generalizable Insights for Graph Transformers in Theory and Practice, Stoll et al., NeurIPS 2025

---

> > ### Author Response · Authors · 2025-11-21
> >
> > >Writing inconsistencies- some datasets have their experiments in main paper, while some have been pushed to appendix.
> >
> > We acknowledge the importance of ensuring key results are accessible within the main paper. Due to space constraints (i.e., page limit) and the breadth of benchmark datasets covered, it was not feasible to include all tables in the main body without sacrificing clarity or omitting critical dataset descriptions and motivations.
> >
> > For the camera-ready version, we will use the additional page to incorporate results from the appendix (e.g., electronic circuits and additional SAT/CO results) into the main text. We have intended to highlight central findings and significant results while maintaining readability and intuition of the importance of each domain.
> >
> > >There are references which may be incorrect and could be a result of LLM-assisted writing or search.
> >
> > To the best of our knowledge, all references cited are correct. All citations were carefully checked for accuracy. If the reviewer can identify any specific incorrect references, we would greatly appreciate clarification so they can be corrected.
> >
> > We hope these clarifications address your concerns. Our primary objective is to provide a solid foundation through diverse data aggregation, accompanied by transparent baseline evaluation, to establish novel benchmarks across different domains.
> >
> > Thank you again for your helpful feedback! Please let us know if you have any remaining concerns. Please consider updating your score if you are satisfied with our responses. We are happy to answer additional questions.

---

### Official Review · Reviewer_JY6e · 2025-10-30

**Soundness:** 3
**Presentation:** 3
**Contribution:** 4
**Rating:** 8
**Confidence:** 3

**Summary:**

This paper introduces GRAPHBENCH, a unified benchmarking suite for graph learning that spans multiple domains (e.g., social networks, chip and circuit design, SAT-solving, combinatorial optimization, and weather forecasting) and supports node-, edge-, graph-level, and generative tasks. The authors argue that current benchmarks are fragmented and overly narrow, and they provide standardized dataset splits, task-relevant metrics, hyperparameter tuning utilities, and baseline comparisons using both MPNNs and graph transformers. The benchmark aims to encourage more realistic evaluation and out-of-distribution generalization.

**Strengths:**

1. The paper is easy to follow and well-written.
2. The paper solves a very important challenge in graph learning, i.e., the limited scope of graph learning.
3. The authors emphasize splits that reflect realistic deployment scenarios (e.g., temporal splits, size-generalization splits).

**Weaknesses:**

1. Although hyperparameter tuning is mentioned as standardized, it is not clear how consistent or fair the tuning budgets are across models. More transparency on tuning methodology and resource constraints would strengthen reproducibility.
2. Some of the datasets, particularly in circuit and combinatorial optimization domains, appear extremely large. Clearer discussion on required computational resources and practical training feasibility would be helpful.

**Questions:**

1. Can the authors provide more details in training and hyperparameter tuning?
2. Can the authors show the computational resource consumption?

---

> ### Author Response · Authors · 2025-11-21
>
> We thank the reviewer for their thoughtful feedback and constructive suggestions. Please find our detailed responses below:
>
> >Although hyperparameter tuning is mentioned as standardized, it is not clear how consistent or fair the tuning budgets are across models. More transparency on tuning methodology and resource constraints would strengthen reproducibility.
>
> We appreciate the reviewer’s emphasis on transparency in hyperparameter tuning methodology and resource constraints.
> In our current submission, we provide an illustrative example of hyperparameter tuning for a selected SAT dataset to demonstrate its impact on results. For all other baseline experiments, due to limited computational resources (in an academic lab), we manually tuned hyperparameters per dataset and task; these settings are fully documented in Appendix D for reproducibility.
>
> Our benchmark is intentionally flexible: users can tune hyperparameters as needed, based on their available resources or research needs. For correct evaluation and meaningful comparison on our benchmark, we recommend that users clearly report their chosen tuning budgets alongside results.
>
> >Question 1: “Can the authors provide more details in training and hyperparameter tuning?”
>
> We manually tuned hyperparameters for each dataset and task (see Appendix D). Because we only provide baseline results—and, as discussed above, feasibility is limited—we do not perform automated (AutoML-based) hyperparameter optimization for every task. However, in Appendix E, we present an example of such tuning for one SAT dataset and show improved results over the manual baseline. We believe this approach balances transparency with practical constraints.
>
> Looking ahead, we agree that standardized automated hyperparameter optimization would further strengthen fairness; in future work or extended releases, we plan to introduce strict budget guidelines for such procedures across tasks where feasible.
>
> >Some of the datasets, particularly in circuit and combinatorial optimization domains, appear extremely large. Clearer discussion on required computational resources and practical training feasibility would be helpful.
>
> Thank you for highlighting concerns about the required computational resources. Our intention is twofold:
>
> First of all, we want to provide challenging datasets that enable the evaluation of scalable graph learning methods on genuinely large datasets, similar to benchmarks such as OGB-LSC. This is in an effort to ensure that GraphBench will not saturate after a short time period but remain a real challenge that pushes forward progress in graph learning.
>
> In addition, we provide some flexibility by having both small- and large-scale versions within each domain so researchers can select data sizes appropriate to their computational resources; it is always possible to train on a subset of the data if needed. We further note that chip design tasks are well known to be hard to solve and require sufficient amounts of data.
>
> To increase transparency around resource requirements, here are exemplary CPU memory usage statistics (RAM) when loading CO datasets entirely into memory:
>
> | Dataset   | RAM (GB) | Runtime
> | ----- | ------- | ------ |
> |RB small |6.69| 35s |
> |ER small |7.02| 35s |
> |BA small |0.75| 19s |
> |RB large |76.55| 4min 43s |
> |ER large |62.95|  3min 58s |
> |BA large |2.06|  30s |
>
> We further provide runtime results for the unsupervised CO experiments (mis), showcasing the resources needed to obtain the results from the paper.
>
> All unsupervised experiments were run on an NVIDIA H100 GPU with 80GB vRAM; supervised experiments were run on an NVIDIA L40S GPU with 48GB vRAM. Runtime results are given for one epoch each.
>
> For all tasks in this release, datasets were loaded fully into RAM; however, users can adapt this approach by loading data incrementally if hardware limitations require it.
>
> >Question 2: Can the authors show the computational resource consumption?
>
> As noted above and in response to Weakness 2, we provide exemplary computational requirements for CO tasks here; these details will be expanded upon in a potential camera-ready version.
>
> In summary, we hope these clarifications address your concerns regarding both hyperparameter transparency/fairness and resource requirements/practical feasibility. Our goal is to support reproducible research while providing flexibility across diverse computational environments.
>
> Thank you again for your valuable feedback! Please let us know if you have any remaining concerns. Please consider updating your score if you are satisfied with our responses. We are happy to answer additional questions.

---

> > ### Comment · Reviewer_JY6e · 2025-11-25
> >
> > Thank you for the response. It resolves my concerns. I think the benchmarking is really a significant issue for the current graph community, and it is a great exploration in this direction. From my perspective, I think the beauty of the work lies in the expansion of graph learning to the "real" real-world problems, like chip design, combinatorial optimization, and even climate forecasting. I will keep my positive rating. I also like to bring two works with similar visions of your work [1,2] and wish they could further inspire your research.
> >
> > [1] Position: Graph Learning Will Lose Relevance Due To Poor Benchmarks, ICML 25.
> >
> > [2] (Section 10) Graph Foundation Models: A Comprehensive Survey.

---

> > > ### Author Response · Authors · 2025-11-25
> > >
> > > Thank you for your positive review. Note that we already cite [1] in the introduction. We will cite [2] in the camera-ready version.

---

### Official Review · Reviewer_DrSB · 2025-11-01

**Soundness:** 2
**Presentation:** 2
**Contribution:** 2
**Rating:** 2
**Confidence:** 4

**Summary:**

This work introduces GraphBench, a benchmarking suite that spans diverse domains and prediction tasks—node-level, edge-level, graph-level, and generative—under standardized evaluation protocols. GraphBench also provides classic GNNs and Graph Transformers as principled baselines to establish reference performance.

**Strengths:**

1. The source code and datasets are provided, enabling reproducibility.
2. The benchmark covers a wide range of domains and prediction tasks.

**Weaknesses:**

The experimental results for the SAT dataset in Table 2 are not informative. The graph-based method (e.g., GIN) is evaluated across different graph representations, while traditional methods are evaluated on hand-crafted features, making the comparison misaligned and difficult to interpret.

Beyond dataset contribution, the **contributions appear limited**.
I recognize that this paper make a good contribution by providing a benchmark aggregating from several new/existing data sources across diverse domains and tasks.
However, as papers in top-tier venues like ICLR generally expect contributions that extend beyond dataset contribution and standardization.
Deeper analysis and substantive insights are needed, e.g., principled evaluations, ablations, and diagnostic studies that advance understanding of when and why certain architectures succeed or fail.

Moreover, the main paper should be **self-contained**. Several key performance metrics are relegated to the appendix, which hinders readability and assessment of the core claims.
For instance, for the CO dataset in Section 3.3.2, the primary performance table appears only as Table 14 in the appendix.

The empirical analysis is shallow.
In Section 3.3.2, the discussion largely describes surface-level performance differences between methods with intuitive but untested rationales tied to model design. I recommend deeper analysis to investigate the performance difference between different methods.

**Questions:**

See weakness.

---

> ### Author Response · Authors · 2025-11-21
>
> We thank the reviewer for their thoughtful feedback and detailed suggestions. We address each point below:
>
> >The experimental results for the SAT dataset in Table 2 are not informative. The graph-based method (e.g., GIN) is evaluated across different graph representations, while traditional methods are evaluated on hand-crafted features, making the comparison misaligned and difficult to interpret.
>
> We respectfully argue that this comparison is not misaligned, but central to the paper's contribution. The goal of the benchmark is to assess the readiness of GNNs and GTs for realistic algorithm selection and performance prediction tasks. Since the current SOTA relies on hand-crafted features, GNNs must be compared against this to assess practical utility. This comparison evaluates the effectiveness of end-to-end learning (GNNs) versus hand-crafted features. Additionally, we believe that GNN-based methods can eventually outperform hand-crafted features, as many of the hand-crafted features are computed over graphs.
>
> >Beyond dataset contribution, the contributions appear limited. I recognize that this paper make a good contribution by providing a benchmark aggregating from several new/existing data sources across diverse domains and tasks. However, as papers in top-tier venues like ICLR generally expect contributions that extend beyond dataset contribution and standardization. Deeper analysis and substantive insights are needed, e.g., principled evaluations, ablations, and diagnostic studies that advance understanding of when and why certain architectures succeed or fail.
>
> We appreciate the reviewer recognizing the value of our benchmark in aggregating diverse data sources across multiple domains and tasks. We respectfully note that, per ICLR’s call for papers, benchmark and dataset contributions are explicitly encouraged as standalone research outputs. The creation, curation, and integration of these datasets represent a very significant undertaking that we believe advances the field by enabling reproducible research and comparison across methods.
>
> While we agree that deeper ablations can yield additional insights, such extensive theoretical or empirical analyses for every baseline architecture across all included domains would be beyond the scope of this work (and feasible in an academic environment). Our focus is to provide a robust foundation upon which such investigations can be built by the graph learning community. Nevertheless, in revision, we will add relevant findings from the existing literature on architectural performance in similar tasks to contextualize our results.
>
> >Moreover, the main paper should be self-contained. Several key performance metrics are relegated to the appendix, which hinders readability and assessment of the core claims. For instance, for the CO dataset in Section 3.3.2, the primary performance table appears only as Table 14 in the appendix.
>
> We acknowledge the importance of ensuring key results are accessible within the main paper. Due to space constraints (i.e., page limit) and the breadth of benchmark datasets covered, it was not feasible to include all tables in the main body without sacrificing clarity or omitting critical dataset descriptions and motivations.
>
> For the camera-ready version, we will use the additional page to move additional core results (e.g., electronic circuits and additional SAT/CO results) into the main text, as suggested. We have intended to highlight central findings and significant results while maintaining readability and intuition of the importance of each domain.

---

> > ### Author Response · Authors · 2025-11-21
> >
> > >The empirical analysis is shallow. In Section 3.3.2, the discussion largely describes surface-level performance differences between methods with intuitive but untested rationales tied to model design. I recommend deeper analysis to investigate the performance difference between different methods.
> >
> > More granular analysis could further illuminate why certain methods succeed or fail on specific datasets. However, given the typical resource constraints in academic labs and given that our primary goal is to enable benchmarking across multiple domains, we prioritized providing solid baselines and an extensible framework over exhaustive empirical study across all settings. The baseline can be seen as a solid starting point.
> >
> > Importantly, our released codebase is designed precisely so researchers can easily evaluate custom models or conduct deeper ablation studies using our infrastructure; we hope this will catalyze future work along these lines. As such, we view extensive analysis as an exciting next step enabled by our contribution rather than its immediate focus.
> >
> >
> > In summary, our work provides substantial value through careful dataset design, rigorous baseline evaluation, and open infrastructure.
> >
> > Thank you again for your constructive comments! Please let us know if you have any remaining concerns. Please consider updating your score if you are satisfied with our responses. We are happy to answer additional questions.

---

### Note · Program_Chairs · 2026-01-17
**Submission Desk Rejected by Program Chairs**

The following references in this submission do not refer to real documents and/or have major errors in bibliographic information:

 V P Dwivedi, A T Luu, and Y Bengio. Graph neural networks with adaptive message passing for long-range dependencies. In Proceedings of Advances in Neural Information Processing Systems NeurIPS, 2022a.
Z Chen, R Sun, and A Gupta. Deepgate 4: Learning logical circuit synthesis with graph transformers. In Proceedings of the International Conference on Learning Representations ICLR, 2025.